# Score-based Pullback Riemannian Geometry: Extracting the Data Manifold Geometry using Anisotropic Flows

**Willem Diepeveen** [* 1]   **Georgios Batzolis** [* 2]   **Zakhar Shumaylov** [3]   **Carola-Bibiane Schönlieb** [3]

## Abstract

Data-driven Riemannian geometry has emerged as a powerful tool for interpretable representation learning, offering improved efficiency in down-stream tasks. Moving forward, it is crucial to balance cheap manifold mappings with efficient training algorithms. In this work, we integrate concepts from pullback Riemannian geometry and generative models to propose a framework for data-driven Riemannian geometry that is scalable in both geometry and learning: score-based pullback Riemannian geometry. Focusing on uni-modal distributions as a first step, we propose a score-based Riemannian structure with closed-form geodesics that pass through the data probability density. With this structure, we construct a Riemannian autoencoder (RAE) with error bounds for discovering the correct data manifold dimension. This framework can naturally be used with anisotropic normalizing flows by adopting isometry regularization during training. Through numerical experiments on diverse datasets, including image data, we demonstrate that the proposed framework produces high-quality geodesics passing through the data support, reliably estimates the intrinsic dimension of the data manifold, and provides a global chart of the manifold. To the best of our knowledge, this is the first scalable framework for extracting the complete geometry of the data manifold.

## 1. Introduction

Data often reside near low-dimensional non-linear manifolds as illustrated in Figure 1. This manifold assumption (Fefferman et al., 2016) has been popular since the early work on non-linear dimension reduction (Belkin & Niyogi, 2001; Coifman & Lafon, 2006; Roweis & Saul, 2000; Sammon, 1969; Tenenbaum et al., 2000). Learning this non-linear structure, or representation learning, from data has proven to be highly successful (DeMers & Cottrell, 1992; Kingma & Welling, 2013) and continues to be a recurring theme in modern machine learning approaches and down-stream applications (Chow et al., 2022; Gomari et al., 2022; Ternes et al., 2022; Vahdat & Kautz, 2020; Zhong et al., 2021).

Recent advances in data-driven Riemannian geometry have demonstrated its suitability for learning representations. In this context, these representations are elements residing in a learned geodesic subspace of the data space, governed by a non-trivial Riemannian structure[1] across the entire ambient space (Arvanitidis et al., 2016; Diepeveen, 2024; Hauberg et al., 2012; Peltonen et al., 2004; Scarvelis & Solomon, 2023; Sorrenson et al., 2024; Sun et al., 2024). Among these contributions, it is worth highlighting that Sorrenson et al. (2024) are the first and only ones so far to use information from the full data distribution obtained though generative models (Dinh et al., 2017; Song et al., 2020), even though this seems a natural approach given recent studies such as Horvat & Pfister (2022); Tempczyk et al. (2022); Sakamoto et al. (2024); Stanczuk et al. (2024). A possible explanation for the limited use of generative models in constructing Riemannian geometry could lie in challenges regarding *scalability of the manifold mappings*. Indeed, even though the generative models can be trained efficiently, Sorrenson et al. (2024) also mention themselves that it can be numerically challenging to work with their induced Riemannian geometry.

If the manifold mapping scalability challenges were to be overcome, the combined power of Riemannian geometry and state of the art generative modelling could have profound implications on how to handle data in general. In-

---

[*]Equal contribution   [1]Department of Mathematics, University of California, Los Angeles, USA [2]Department of Engineering, University of Cambridge, UK [3]Faculty of Mathematics, University of Cambridge, UK. Correspondence to: Willem Diepeveen <wdiepeveen@math.ucla.edu>, Georgios Batzolis <gb511@cam.ac.uk>.

*Proceedings of the $42^{nd}$ International Conference on Machine Learning*, Vancouver, Canada. PMLR 267, 2025. Copyright 2025 by the author(s).

---

[1]rather than the standard $\ell^2$-inner product

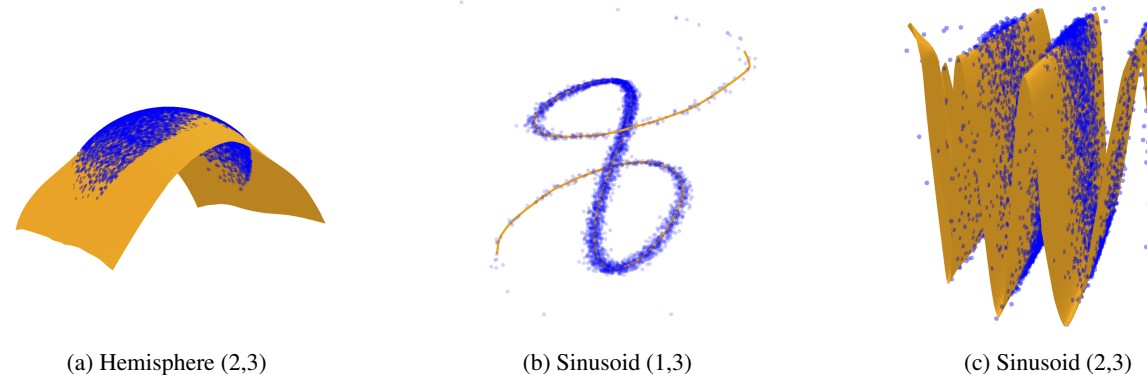

|  |  |  |
|---|---|---|
| (a) Hemisphere (2,3) | (b) Sinusoid (1,3) | (c) Sinusoid (2,3) |

*Figure 1.* Approximate data manifolds learned by the Riemannian autoencoder generated by score-based pullback Riemannian geometry for three datasets. The orange surfaces represent the manifolds learned by the model, while the blue points correspond to the training data. Each manifold provides a convincing low-dimensional representation of the data, isometric to its respective latent space.

deed, beyond typical data analysis tasks such as computing distances, means, and interpolations/extrapolations of data points, a data-driven Riemannian structure also offers greater potential for representation learning and downstream applications. For instance, many advanced data processing methods, from Principal Component Analysis (PCA) to score and flow-matching, have Riemannian counterparts (Diepeveen et al. (2023); Fletcher et al. (2004) and Chen & Lipman (2023); Huang et al. (2022)) that have proven beneficial by improving upon full black box methods in terms of interpretability (Diepeveen, 2024) or Euclidean counterparts in terms of efficiency (Kapusniak et al., 2024; de Kruiff et al., 2024). Here it is worth highlighting that scalability of manifold mappings was completely circumvented by (Diepeveen, 2024) and (de Kruiff et al., 2024) by using pullback geometry. However, here learning a suitable (and stable) pullback geometry suffers from challenges regarding *scalability of the training algorithm*, contrary to the approach by (Sorrenson et al., 2024).

Motivated by the above, this work aims to address the following question: How to strike a good balance between scalability of training a data-driven Riemannian structure and of evaluating its corresponding manifold mappings?

### 1.1. Contributions

In this paper, we take first steps towards striking such a balance and propose a score-based pullback Riemannian metric assuming a relatively simple but generally applicable family of probability densities, which we show to result in both scalable manifold mappings and scalable learning algorithms. We emphasize that we do not directly aim to find the perfect balance between the two types of scalability. Instead we start from a setting which has many nice properties, but will allow for generalization to multimodal densities, which we reserve for future work.

Specifically, we consider a family of unimodal probability densities whose negative log-likelihoods are compositions of strongly convex functions and diffeomorphisms. As this work is an attempt to bridge between the geometric data analysis community and the generative modeling community, we break down the contributions in two ways. Theoretically,

- We propose a score-based pullback Riemannian metric such that manifold mappings respect the data distribution.

- We demonstrate that this density-based Riemannian structure naturally leads to a Riemannian autoencoder[2] and provide error bounds on the expected reconstruction error, which allows for approximation of the data manifold as illustrated in Figure 1.

- We introduce a learning scheme based on adaptations of normalizing flows to find the density to be integrated into the Riemannian framework, which is tested on several synthetic data sets.

Practically, this work showcases how two simple adaptations to the normalizing flows framework enable data-driven Riemannian geometry. This significantly expands the potential for downstream applications compared to the unadapted framework.

### 1.2. Outline

After introducing notation in Section 2, Section 3 considers a family of probability distributions, from which we obtain suitable geometry, and Section 4 showcases how one can subsequently construct Riemannian Autoencoders with theoretical guarantees. From these observations Section 5 discusses the natural limitations of standard normalizing

---

[2]in the sense of Diepeveen (2024)

flows and how to change the parametrisation and training for downstream application in a Riemannian geometric setting. Section 6 showcases several use cases of data-driven Riemannian structure on several data sets. Finally, we summarize our findings in Section 7.

## 2. Notation

Here we present some basic notations from differential and Riemannian geometry, see Boothby (2003); Carmo (1992); Lee (2013); Sakai (1996) for details.

*Smooth Manifolds and Tangent Spaces.* Let $\mathcal{M}$ be a smooth manifold. We write $C^\infty(\mathcal{M})$ for the space of smooth functions over $\mathcal{M}$. The *tangent space* at $\mathbf{p} \in \mathcal{M}$, which is defined as the space of all *derivations* at $\mathbf{p}$, is denoted by $\mathcal{T}_\mathbf{p}\mathcal{M}$ and for *tangent vectors* we write $\Xi_\mathbf{p} \in \mathcal{T}_\mathbf{p}\mathcal{M}$. For the *tangent bundle* we write $\mathcal{T}\mathcal{M}$ and smooth vector fields, which are defined as *smooth sections* of the tangent bundle, are written as $\mathscr{X}(\mathcal{M}) \subset \mathcal{T}\mathcal{M}$.

*Riemannian Manifolds.* A smooth manifold $\mathcal{M}$ becomes a *Riemannian manifold* if it is equipped with a smoothly varying *metric tensor field* $(\cdot, \cdot) \colon \mathscr{X}(\mathcal{M}) \times \mathscr{X}(\mathcal{M}) \to C^\infty(\mathcal{M})$. This tensor field induces a *(Riemannian) metric* $d_\mathcal{M} \colon \mathcal{M} \times \mathcal{M} \to \mathbb{R}$. The metric tensor can also be used to construct a unique affine connection, the *Levi-Civita connection*, that is denoted by $\nabla_{(\cdot)}(\cdot) \colon \mathscr{X}(\mathcal{M}) \times \mathscr{X}(\mathcal{M}) \to \mathscr{X}(\mathcal{M})$. This connection is in turn the cornerstone of a myriad of manifold mappings.

*Geodesic.* One is the notion of a *geodesic*, which for two points $\mathbf{p}, \mathbf{q} \in \mathcal{M}$ is defined as a curve $\gamma_{\mathbf{p},\mathbf{q}} \colon [0,1] \to \mathcal{M}$ with minimal length that connects $\mathbf{p}$ with $\mathbf{q}$. Another closely related notion to geodesics is the curve $t \mapsto \gamma_{\mathbf{p},\Xi_\mathbf{p}}(t)$ for a geodesic starting from $\mathbf{p} \in \mathcal{M}$ with velocity $\dot{\gamma}_{\mathbf{p},\Xi_\mathbf{p}}(0) = \Xi_\mathbf{p} \in \mathcal{T}_\mathbf{p}\mathcal{M}$.

*Exponential Map.* This can be used to define the *exponential map* $\exp_\mathbf{p} \colon \mathcal{D}_\mathbf{p} \to \mathcal{M}$ as $\exp_\mathbf{p}(\Xi_\mathbf{p}) := \gamma_{\mathbf{p},\Xi_\mathbf{p}}(1)$, where $\mathcal{D}_\mathbf{p} \subset \mathcal{T}_\mathbf{p}\mathcal{M}$ is the set on which $\gamma_{\mathbf{p},\Xi_\mathbf{p}}(1)$ is defined.

*Logarithmic Map.* Furthermore, the *logarithmic map* $\log_\mathbf{p} \colon \exp(\mathcal{D}'_\mathbf{p}) \to \mathcal{D}'_\mathbf{p}$ is defined as the inverse of $\exp_\mathbf{p}$, so it is well-defined on $\mathcal{D}'_\mathbf{p} \subset \mathcal{D}_\mathbf{p}$ where $\exp_\mathbf{p}$ is a diffeomorphism.

*Pullback metrics.* Finally, if $(\mathcal{M}, (\cdot, \cdot))$ is a $d$-dimensional Riemannian manifold, $\mathcal{N}$ is a $d$-dimensional smooth manifold and $\phi \colon \mathcal{N} \to \mathcal{M}$ is a diffeomorphism, the *pullback metric*

$$(\Xi, \Phi)^\phi := (D_{(\cdot)}\phi[\Xi_{(\cdot)}], D_{(\cdot)}\phi[\Phi_{(\cdot)}])_{\phi(\cdot)}, \quad (1)$$

where $D_\mathbf{p}\phi \colon \mathcal{T}_\mathbf{p}\mathcal{N} \to \mathcal{T}_{\phi(\mathbf{p})}\mathcal{M}$ denotes the differential of $\phi$, defines a Riemannian structure on $\mathcal{N}$, which we denote by $(\mathcal{N}, (\cdot, \cdot)^\phi)$. Pullback metrics literally pull back all geometric information from the Riemannian structure on $\mathcal{M}$.

Throughout the rest of the paper pullback mappings will be denoted similarly to Equation (1) with the diffeomorphism $\phi$ as a superscript, i.e., we write $d_\mathcal{N}^\phi(\mathbf{p}, \mathbf{q})$, $\gamma_{\mathbf{p},\mathbf{q}}^\phi$, $\exp_\mathbf{p}^\phi(\Xi_\mathbf{p})$ and $\log_\mathbf{p}^\phi \mathbf{q}$ for $\mathbf{p}, \mathbf{q} \in \mathcal{N}$ and $\Xi_\mathbf{p} \in \mathcal{T}_\mathbf{p}\mathcal{N}$.

## 3. Riemannian geometry from unimodal probability densities

We remind the reader that the ultimate goal of data-driven Riemannian geometry on $\mathbb{R}^d$ is to construct a Riemannian structure such that geodesics always pass through the support of data probability densities. In this section we will focus on constructing Riemannian geometry that does just that from unimodal densities $p \colon \mathbb{R}^d \to \mathbb{R}$ of the form

$$p(\mathbf{x}) \propto e^{-\psi(\varphi(\mathbf{x}))}, \quad (2)$$

where $\psi \colon \mathbb{R}^d \to \mathbb{R}$ is a smooth strongly convex function and $\varphi \colon \mathbb{R}^d \to \mathbb{R}^d$ is a diffeomorphism. In particular, we will consider pullback Riemannian structures of the form[3]

$$(\Xi, \Phi)_\mathbf{x}^{\nabla\psi\circ\varphi} := (D_\mathbf{x}\nabla\psi \circ \varphi[\Xi], D_\mathbf{x}\nabla\psi \circ \varphi[\Phi])_2. \quad (3)$$

For proofs of the results below and those of more general statements we refer the reader to Appendix A.

The following result, which is a direct application of (Diepeveen, 2024, Prop. 2.1), gives us closed-form expressions of several important manifold mappings under $(\cdot, \cdot)^{\nabla\psi\circ\varphi}$ if we choose

$$\psi(\mathbf{x}) = \frac{1}{2}\mathbf{x}^\top \mathbf{A}^{-1}\mathbf{x}, \quad (4)$$

where $\mathbf{A} \in \mathbb{R}^{d\times d}$ is symmetric positive definite.

**Proposition 3.1.** *Let $\varphi \colon \mathbb{R}^d \to \mathbb{R}^d$ be a smooth diffeomorphism and let $\psi \colon \mathbb{R}^d \to \mathbb{R}$ be a function of the form Equation (4).*

*Then,*

$$d_{\mathbb{R}^d}^{\nabla\psi\circ\varphi}(\mathbf{x}, \mathbf{y}) = \|\mathbf{A}^{-1}(\varphi(\mathbf{x}) - \varphi(\mathbf{y}))\|_2, \quad (5)$$

$$\gamma_{\mathbf{x},\mathbf{y}}^{\nabla\psi\circ\varphi}(t) = \varphi^{-1}((1-t)\varphi(\mathbf{x}) + t\varphi(\mathbf{y})), \quad (6)$$

$$\exp_\mathbf{x}^{\nabla\psi\circ\varphi}(\Xi_\mathbf{x}) = \varphi^{-1}(\varphi(\mathbf{x}) + D_\mathbf{x}\varphi[\Xi_\mathbf{x}]), \quad (7)$$

$$\log_\mathbf{x}^{\nabla\psi\circ\varphi} \mathbf{y} = D_{\varphi(\mathbf{x})}\varphi^{-1}[\varphi(\mathbf{y}) - \varphi(\mathbf{x})]. \quad (8)$$

*Remark* 3.2. We note that $\ell^2$-stability of geodesics is inherited by (Diepeveen, 2024, Thm. 3.4), if we have (approximate) local $\ell^2$-isometry of $\varphi$ on the data support.

A direct result of Proposition 3.1 is that geodesics will pass through the support of $p(\mathbf{x})$ from (2), in the sense that geodesics pass through regions with higher likelihood than the end points. This can be formalized in the following result.

---

[3]Note that $\nabla\psi \circ \varphi$ should be read as $(\nabla\psi) \circ \varphi$.

**Theorem 3.3.** *Let $\varphi : \mathbb{R}^d \to \mathbb{R}^d$ be a smooth diffeomorphism and let $\psi : \mathbb{R}^d \to \mathbb{R}$ be a function of the form (4).*

*Then, mapping*

$$t \mapsto \psi(\varphi(\gamma_{\mathbf{x},\mathbf{y}}^{\nabla \psi \circ \varphi}(t))), \quad t \in [0,1] \qquad (9)$$

*is strongly convex.*

The Riemannian structure in Equation (3) is related to the Riemannian structure obtained from the *score function*[4] $\nabla \log(p(\cdot)) : \mathbb{R}^d \to \mathbb{R}^d$ if $\varphi$ is close to a smooth local $\ell^2$-isometry on the data support, i.e., $D_{\mathbf{x}}\varphi$ is an orthogonal operator:

$$(D_{\mathbf{x}}\nabla \log(p(\cdot))[\Xi], D_{\mathbf{x}}\nabla \log(p(\cdot))[\Phi])_2$$
$$= (D_{\mathbf{x}}\nabla(\psi \circ \varphi)[\Xi], D_{\mathbf{x}}\nabla(\psi \circ \varphi)[\Phi])_2$$
$$= (D_{\mathbf{x}}((D_{(\cdot)}\varphi)^\top \circ \nabla \psi \circ \varphi)[\Xi],$$
$$\qquad D_{\mathbf{x}}\left((D_{(\cdot)}\varphi)^\top \circ \nabla \psi \circ \varphi\right)[\Phi])_2$$
$$\approx (D_{\mathbf{x}}\nabla \psi \circ \varphi[\Xi], D_{\mathbf{x}}\nabla \psi \circ \varphi[\Phi])_2 = (\Xi, \Phi)_{\mathbf{x}}^{\nabla \psi \circ \varphi}. \quad (10)$$

For that reason, we call such an approach to data-driven Riemannian geometry: *score-based pullback Riemannian geometry*.

## 4. Riemannian autoencoder from unimodal probability densities

Proposition 3.1 begs the question what $\psi$ could still be used for. We note that this case comes down to having a data probability density that is a deformed Gaussian distribution. In the case of a regular (non-deformed) Gaussian, one can compress the data generated by it through projecting them onto a low rank approximation of the covariance matrix such that only the directions with highest variance are taken into account. This is the basic idea behind PCA. In the following we will generalize this idea to the Riemannian setting and observe that this amounts to constructing a *Riemannian autoencoder* (RAE) (Diepeveen, 2024, Sec. 4), whose error we can bound by picking the dimension of the autoencoder in a clever way, reminiscent of the classical PCA error bound.

Concretely, we assume that we have a unimodal density of the form Equation (2) with a quadratic strongly convex function $\psi(\mathbf{x}) := \frac{1}{2}\mathbf{x}^\top \mathbf{A}^{-1}\mathbf{x}$ for some diagonal matrix $\mathbf{A} := \operatorname{diag}(\mathbf{a}_1, \ldots \mathbf{a}_d)$ with positive entries[5]. Next, we define an indexing $u_w \in [d] := \{1, \ldots, d\}$ for $w = 1, \ldots, d$ such that

$$\mathbf{a}_{u_1} \geq \ldots \geq \mathbf{a}_{u_d}, \qquad (11)$$

---

[4]Note that the score by itself is not always a diffeomorphism.

[5]Note that this is not restrictive as for a general symmetric positive definite matrix $\mathbf{A}$ the eigenvalues can be used as diagonal entries and the orthonormal matrices can be concatenated with the diffeomorphism.

and consider a threshold $\varepsilon \in [0,1]$. We then consider $d_\varepsilon \in [d]$ defined as the integer that satisfies

$$d_\varepsilon = \min\left\{ d' \in [d-1] \;\Big|\; \sum_{w=d'+1}^{d} \mathbf{a}_{u_w} \leq \varepsilon \sum_{u=1}^{d} \mathbf{a}_u \right\} \quad (12)$$

if $\mathbf{a}_{u_d} \leq \varepsilon \sum_{u=1}^{d} \mathbf{a}_u$ and $d_\varepsilon = d$ otherwise.

Finally, we define the encoder (chart) $E_\varepsilon : \mathbb{R}^d \to \mathbb{R}^{d_\varepsilon}$

$$E_\varepsilon(\mathbf{x})_w := (\log_{\varphi^{-1}(\mathbf{0})}^\varphi \mathbf{x}, D_{\mathbf{0}}\varphi^{-1}[\mathbf{e}^{u_w}])_{\varphi^{-1}(\mathbf{0})}^\varphi$$
$$\overset{\text{Equation (8)}}{=} (\varphi(\mathbf{x}), \mathbf{e}^{u_w})_2, \quad w = 1, \ldots, d_\varepsilon, \quad (13)$$

and define the decoder (inverse chart) $D_\varepsilon : \mathbb{R}^{d_\varepsilon} \to \mathbb{R}^d$ as

$$D_\varepsilon(\mathbf{p}) := \exp_{\varphi^{-1}(\mathbf{0})}^\varphi \left( \sum_{w=1}^{d_\varepsilon} \mathbf{p}_w D_{\mathbf{0}}\varphi^{-1}[\mathbf{e}^{u_w}] \right)$$
$$\overset{\text{Equation (7)}}{=} \varphi^{-1}\left( \sum_{w=1}^{d_\varepsilon} \mathbf{p}_w \mathbf{e}^{u_w} \right), \quad (14)$$

which generate a Riemannian autoencoder and the set $D_\varepsilon(\mathbb{R}^{d_\varepsilon}) \subset \mathbb{R}^d$ as an approximate data manifold as in the scenario in Figure 1.

Similarly to classical PCA, this Riemannian autoencoder comes with an error bound on the expected approximation error, which is fully determined by the diffeomorphism's deviation from isometry around the data manifold. For the proof, we refer the reader to Appendix B.

**Theorem 4.1.** *Let $\varphi : \mathbb{R}^d \to \mathbb{R}^d$ be a smooth diffeomorphism and let $\psi : \mathbb{R}^d \to \mathbb{R}$ be a quadratic function of the form Equation (4) with positive definite diagonal matrix $\mathbf{A} \in \mathbb{R}^{d \times d}$. Furthermore, let $p : \mathbb{R}^d \to \mathbb{R}$ be the corresponding probability density of the form Equation (2). Finally, consider $\varepsilon \in [0,1]$ and the mappings $E_\varepsilon : \mathbb{R}^d \to \mathbb{R}^{d_\varepsilon}$ and $D_\varepsilon : \mathbb{R}^{d_\varepsilon} \to \mathbb{R}^d$ in Equations (13) and (14) with $d_\varepsilon \in [d]$ as in Equation (12).*

*Then,*

$$\mathbb{E}_{\mathbf{X} \sim p}[\|D_\varepsilon(E_\varepsilon(\mathbf{X})) - \mathbf{X}\|_2^2] \leq \varepsilon C_\varphi \operatorname{tr}(\mathbf{A}) + o(\varepsilon), \quad (15)$$

*where*

$$C_\varphi := \inf_{\beta \in [0,\frac{1}{2})} \left\{ \frac{C_{\beta,\varphi}^1 C_{\beta,\varphi}^2 C_{\beta,\varphi}^3}{1 - 2\beta} \left( \frac{1+\beta}{1-2\beta} \right)^{\frac{d}{2}} \right\} \quad (16)$$

*for*

$$C_{\beta,\varphi}^1 := \sup_{\mathbf{x} \in \mathbb{R}^d} \left\{ \|D_{\varphi(\mathbf{x})}\varphi^{-1}\|_2^2 e^{-\frac{\beta}{2}\varphi(\mathbf{x})^\top \mathbf{A}^{-1}\varphi(\mathbf{x})} \right\}, \quad (17)$$

$$C_{\beta,\varphi}^2 := \sup_{\mathbf{x} \in \mathbb{R}^d} \left\{ |\det(D_{\mathbf{x}}\varphi)| e^{-\frac{\beta}{2}\varphi(\mathbf{x})^\top \mathbf{A}^{-1}\varphi(\mathbf{x})} \right\}, \quad (18)$$

*and*

$$C_{\beta,\varphi}^3 := \sup_{\mathbf{x} \in \mathbb{R}^d} \{|\det(D_{\varphi(\mathbf{x})}\varphi^{-1})|e^{-\frac{\beta}{2}\varphi(\mathbf{x})^\top \mathbf{A}^{-1}\varphi(\mathbf{x})}\}.$$

(19)

*Remark* 4.2. Note that the RAE latent space is interpretable as it is $\ell^2$-isometric to the data manifold if $\varphi$ is an approximate $\ell^2$-isometry on the data manifold. In other words, latent representations being close by or far away correspond to similar behaviour in data space, which is not the case for a VAE (Kingma & Welling, 2013).

## 5. Learning unimodal probability densities

Naturally, we want to learn probability densities of the form Equation (2), which can then directly be inserted into the proposed score-based pullback Riemannian geometry framework. In this section we will consider how to adapt normalizing flow (NF) (Dinh et al., 2017) training to a setting that is more suitable for our purposes[6]. In particular, we will consider how training a normalizing flow density $p : \mathbb{R}^d \to \mathbb{R}$ given by

$$p(\mathbf{x}) := \frac{1}{C_\psi} e^{-\psi(\varphi(\mathbf{x}))} |\det(D_{\mathbf{x}}\varphi)|,$$

(20)

where $C_\psi > 0$ is a normalisation constant that only depends on the strongly convex function $\psi$, yields our target distribution Equation (2).

From Sections 3 and 4 we have seen that ideally the strongly convex function $\psi : \mathbb{R}^d \to \mathbb{R}$ corresponds to a Gaussian with a parameterised diagonal covariance matrix $\mathbf{A} \in \mathbb{R}^{d \times d}$, resulting in more parameters than in standard normalizing flows, whereas the diffeomorphism $\varphi : \mathbb{R}^d \to \mathbb{R}^d$ is regularized to be an isometry. In particular, $\mathbf{A}$ ideally allows for learnable anisotropy rather than having a fixed isotropic identity matrix. The main reason is that through anisotropy we can construct a Riemannian autoencoder (RAE), since it is known which dimensions are most important. Moreover, the diffeomorphism should be $\ell^2$-isometric, unlike standard normalizing flows which are typically non-volume preserving, enabling stability (Remark 3.2) and a practically useful and interpretable RAE (Theorem 4.1 and Remark 4.2).

In addition, $\ell^2$-isometry (on the data support) implies volume-preservation, which means that $|\det(D_{\mathbf{x}}\varphi)| \approx 1$ so that the model density (20) reduces to the desired form of Equation (2)[7]. While volume preservation theoretically follows from $\ell^2$-isometry, in practice, the flow can only approximate local isometry through optimization. Thus, we found it beneficial to explicitly include a volume-preservation loss, resulting in an adapted normalizing flow loss that enforces both constraints for improved alignment with the desired Riemannian structure.

$$\mathcal{L}(\theta_1, \theta_2) := \mathbb{E}_{\mathbf{X} \sim p_{\text{data}}} \left[ -\log p_{\theta_1, \theta_2}(\mathbf{X}) \right]$$
$$+ \lambda_{\text{vol}} \mathbb{E}_{\mathbf{X} \sim p_{\text{data}}} \left[ \log(|\det(D_{\mathbf{X}}\varphi_{\theta_2})|)^2 \right]$$
$$+ \lambda_{\text{iso}} \mathbb{E}_{\mathbf{X} \sim p_{\text{data}}} \left[ \|(D_{\mathbf{X}}\varphi_{\theta_2})^\top D_{\mathbf{X}}\varphi_{\theta_2} - \mathbf{I}_d\|_F^2 \right] \quad (21)$$

where $\lambda_{\text{vol}}, \lambda_{\text{iso}} > 0$ and the negative log likelihood term reduces to

$$\frac{1}{2} \mathbb{E}_{\mathbf{X} \sim p_{\text{data}}} \left[ \varphi_{\theta_2}(\mathbf{X})^\top \mathbf{A}_{\theta_1}^{-1} \varphi_{\theta_2}(\mathbf{X}) \right] + \frac{1}{2} \text{tr}(\log(\mathbf{A}_{\theta_1}))$$
$$- \mathbb{E}_{\mathbf{X} \sim p_{\text{data}}} \left[ \log(|\det(D_{\mathbf{X}}\varphi_{\theta_2})|) \right] + \frac{d}{2} \log(2\pi), \quad (22)$$

where $\mathbf{A}_{\theta_1}$ is a diagonal matrix and $\varphi_{\theta_2}$ is a normalizing flow with affine coupling layers[8] (Dinh et al., 2017). For small ambient dimensions, isometry regularization is feasible, but in high dimensions, computing the full Jacobian is impractical. To address this, we use a scalable sliced isometry loss based on Jacobian-vector products, significantly reducing both computational and memory costs while preserving effectiveness. See Appendix I.2 for details.

## 6. Experiments

We conducted two sets of experiments to evaluate the proposed scheme from 5 to learn suitable pullback Riemannian geometry. The first set investigates whether our adaptation of the standard normalizing flow (NF) training paradigm leads to more accurate and stable manifold mappings, as measured by the geodesic and variation errors. The second set assesses the capability of our method to generate a robust Riemannian autoencoder (RAE).

For all experiments in this section, detailed training configurations are provided in Appendix C and additional results in Appendix D.

### 6.1. Manifold mappings

As discussed in (Diepeveen, 2024), the quality of learned manifold mappings is determined by two key metrics: the *geodesic error* and the *variation error*. The geodesic error measures the average deviation form the ground truth

---

[6]We note that the choice for adapting the normalizing flow training scheme rather than using diffusion model training schemes is due to more robust results through the former.

[7]We note that without these constraints (accommodating multimodality) the learned mappings can in principle be used to construct Riemannian geometry and a RAE. However, from the theory discussed in this paper we cannot guarantee stability of manifold mappings nor that the RAE has the right dimension.

[8]We note that the choice for affine coupling layers rather than using more expressive diffeomorphisms such as rational quadratic flows (Durkan et al., 2019) is due to our need for high regularity for stable manifold mappings (Remark 3.2) and an interpretable RAE (Remark 4.2), which has empirically shown to be more challenging to achieve for more expressive flows as both first-and higher-order derivatives of $\varphi$ will blow up the error terms in Theorem 4.1.

geodesics implied by the ground truth pullback metric, while the variation error evaluates the stability of geodesics under small perturbations. We define these error metrics for the evaluation of pullback geometries in Appendix E.

Our approach introduces two key modifications to the normalizing flow (NF) training framework:

1. **Anisotropic Base Distribution**: We parameterize the diagonal elements of the covariance matrix $\mathbf{A}_{\theta_1}$, introducing anisotropy into the base distribution.

2. $\ell^2$**-Isometry Regularization**: We regularize the flow $\varphi_{\theta_2}$ to be approximately $\ell^2$-isometric.

To assess the effectiveness of these modifications in learning more accurate and robust manifold mappings, we compare our method against three baselines:

(1) *Normalizing Flow (NF)*: Standard NF with an isotropic Gaussian base $\mathcal{N}(\mathbf{0}, \mathbf{I}_d)$, no isometry regularization.

(2) *Anisotropic NF*: NF with a parameterized diagonal covariance base, no isometry regularization.

(3) *Isometric NF*: NF with an isotropic Gaussian base, regularized to be $\ell^2$-isometric.

We conduct experiments on three datasets, illustrated in 7 in F.1: the *Single Banana Dataset*, the *Squeezed Single Banana Dataset*, and the *River Dataset*. Detailed descriptions of the construction and characteristics of these datasets are provided in F.1.

Table 1 presents the geodesic and variation errors for each method across the three datasets and Figure 2 visually compares the geodesics computed using each method on the river dataset. Our method consistently achieves significantly lower errors compared to the baselines, indicating more accurate and stable manifold mappings.

Introducing anisotropy in the base distribution without enforcing isometry in the flow offers no significant improvement over the standard flow. On the other hand, regularizing the flow to be approximately isometric without incorporating anisotropy in the base distribution results in underfitting, leading to noticeably worse performance than the standard flow. Our results demonstrate that the combination of anisotropy in the base distribution with isometry regularization (our method) yields the most accurate and stable manifold mappings, as evidenced by consistently lower geodesic and variation errors.

### 6.2. Riemannian Autoencoder

To evaluate our method's ability to learn a Riemannian autoencoder (RAE), we conducted experiments on both low-to-moderate dimensional *Euclidean* datasets and higher-dimensional *image* datasets:

- *Hemisphere*($d'$, $d$) and *Sinusoid*($d'$, $d$): Synthetic Euclidean datasets with controllable intrinsic dimension $d'$ and ambient dimension $d$.

- $d'$-*Gaussian Blobs Image Manifold*: A synthetic image dataset, also with controllable intrinsic dimension $d'$, embedded in a 1024-dimensional ambient space.

- *MNIST*: A dataset of handwritten digits, originally embedded in a 784-dimensional space, which we further embed into a 1024-dimensional ambient space using bicubic rescaling.

The *Hemisphere* and *Sinusoid* datasets are used to evaluate RAE's performance on low-to-moderate dimensional manifolds, while the *Gaussian Blobs* and *MNIST* datasets are used to test its scalability to higher-dimensional and more complex image manifolds. For further details on each dataset, refer to Appendix F.2.

#### 6.2.1. EUCLIDEAN DATASETS

**1D and 2D manifolds.** In Figures 1 and 5, we present the data manifold approximations by our Riemannian autoencoder for four low-dimensional manifolds: Hemisphere(2,3), Sinusoid(1,3), Sinusoid(2,3) and Sinusoid(1,100). In Appendix G, we detail the process used to create the data manifold approximations for these experiments. In our experiments, we set $\epsilon = 0.01$, which resulted in $d_\epsilon = d'$ for all cases, accurately capturing the intrinsic dimension of each manifold and producing accurate global charts.

**Higher-dimensional manifolds.** To assess the scalability of our method, we conducted experiments on the Hemisphere(5,20) and Sinusoid(5,20) datasets. The learned variances effectively indicate the importance of each latent dimension, with high variances corresponding to the intrinsic manifold structure.

On the Hemisphere(5,20) dataset, our model correctly identified five non-vanishing latent dimensions, achieving near-zero reconstruction error when selecting them. In contrast, choosing latent dimensions with vanishing variance resulted in no meaningful error reduction, confirming the model's ability to separate important from redundant dimensions. A more detailed analysis of this effect is provided in Appendix D.

For the more challenging Sinusoid(5,20) dataset, our method remains highly effective, though slightly less precise than for the Hemisphere dataset. The first six most important latent dimensions account for approximately 97% of the

| Metric | Our Method | NF | Anisotropic NF | Isometric NF |
|---|---|---|---|---|
| **Single Banana Dataset** | | | | |
| Geodesic Error | **0.0315 (0.0268)** | 0.0406 (0.0288) | 0.0431 (0.0305) | 0.0817 (0.1063) |
| Variation Error | **0.0625 (0.0337)** | 0.0638 (0.0352) | 0.0639 (0.0354) | 0.0639 (0.0355) |
| **Squeezed Single Banana Dataset** | | | | |
| Geodesic Error | **0.0180 (0.0226)** | 0.0524 (0.0805) | 0.0505 (0.0787) | 0.1967 (0.2457) |
| Variation Error | **0.0631 (0.0326)** | 0.0663 (0.0353) | 0.0661 (0.0350) | 0.0669 (0.0361) |
| **River Dataset** | | | | |
| Geodesic Error | **0.1691 (0.0978)** | 0.2369 (0.1216) | 0.2561 (0.1338) | 0.3859 (0.2568) |
| Variation Error | **0.0763 (0.0486)** | 0.1064 (0.0807) | 0.1113 (0.0863) | 0.0636 (0.0333) |

*Table 1.* Comparison of evaluation metrics for different methods across three datasets. Best-performing results for each metric are highlighted in bold. Values are reported as mean (std). The proposed method performs best in all metrics on each data set.

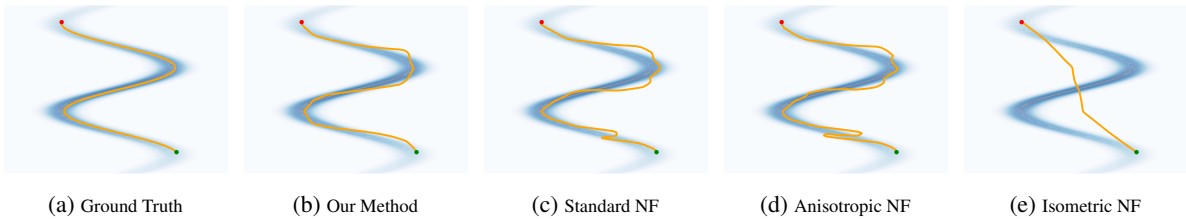

(a) Ground Truth     (b) Our Method     (c) Standard NF     (d) Anisotropic NF     (e) Isometric NF

*Figure 2.* Comparison of geodesics computed using different methods on the river dataset. The geodesics generated by the proposed method have the fewest artifacts, aligning with expectations from Table 1.

variance, increasing to over 99% with the seventh dimension. The model slightly overestimates the intrinsic dimensionality, likely due to the increased optimization challenges of learning a more intricate distribution while regularizing for isometry.

### 6.2.2. IMAGE DATASETS

In our preliminary experiments on image datasets, we observed significant overestimation of the intrinsic dimension, despite the high quality of the geodesic interpolations. This issue stems from the use of non-affine flows,[9] which increase the model's flexibility but also inflate a Hessian-vector product term in the expansion of the Hessian of the model log-density. This inflation can disrupt the RAE's ability to assign high variances to the most important latent dimensions (see Appendix H for details).

To address this issue, we repeated the image experiments after including the Hessian vector product term as an additional regularization term in the loss function. The additional regularization term is given by:

$$\lambda_{\text{hess}} \, \mathbb{E}_{\mathbf{X} \sim p_{\text{data}}} \left[ \left\| D^2_{\mathbf{X}} \varphi_{\theta_2} \cdot \mathbf{A}^{-1}_{\theta_1} \varphi_{\theta_2}(\mathbf{X}) \right\|_2 \right], \quad (23)$$

where $\lambda_{\text{hess}}$ is the regularization weight, and $D^2_{\mathbf{X}} \varphi_{\theta_2}$ denotes the Hessian of the flow. As demonstrated in the following

experiments, introducing this term substantially improved the RAE's ability to correctly detect the intrinsic dimension while maintaining smooth geodesic paths and accurate reconstructions.

Computing the Hessian-vector product is impractical in high dimensions. To improve scalability, we instead minimize the norm of a single randomly chosen column of the Hessian-vector product ($\mathbb{R}^{d \times d}$) per iteration. This significantly reduces computational and memory overhead while preserving effectiveness. See Appendix I.3 for details.

**$d'$-Gaussian Blobs.** We evaluated our RAE on synthetic $d'$-*Gaussian Blobs* manifolds embedded in a 1024-dimensional ambient space with intrinsic dimensions $d' = 20, 50,$ and $100$. The RAE accurately captured the structure, assigning over 99% of the total variance to 22, 51, and 101 latent dimensions respectively, closely aligning with the ground-truth intrinsic dimensions.

Figure 3 shows the normalized cumulative variance and the $L^2$ reconstruction error as functions of the number of most important latent dimensions. All curves saturate very close to the ground truth intrinsic dimension demonstrating RAE's ability to effectively capture the intrinsic structure of the data in high dimensional settings.

**MNIST.** We evaluated the RAE on the MNIST dataset, a real-world benchmark known for its multimodal distribution (each digit class represents a separate "mode"). While our

---

[9]We used compositions of affine coupling layers and $1 \times 1$ invertible convolutions, with the latter introducing non-affine transformations to the flow.

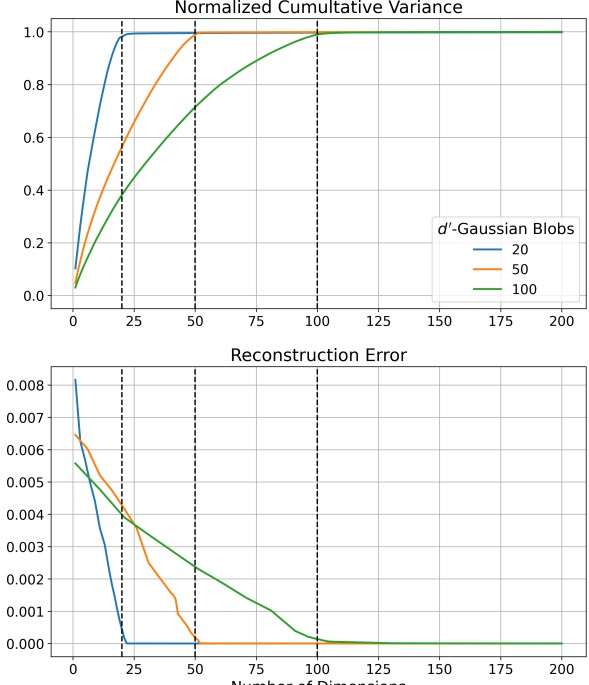

*Figure 3.* Normalized cumulative variance (top) and $L^2$ reconstruction error (bottom) for $d'$-Gaussian Blobs with intrinsic dimensions $d' = 20, 50,$ and $100$. Black dotted vertical lines mark the ground-truth intrinsic dimensions for each dataset. Curves are shown for the first 200 dimensions to highlight the critical range, as the behavior from 200 to 1024 dimensions remains effectively constant.

method is theoretically best suited for unimodal distributions of the form in Equation (2), it still performs remarkably well on MNIST, with only a slight tendency to overestimate the intrinsic dimension. Specifically, at $\epsilon = 0.1$, the RAE assigns 90% of the variance to 176 latent dimensions, which slightly exceeds the maximum local intrinsic dimension (LID) of 152 estimated by ID-DIFF, a state-of-the-art LID method (Stanczuk et al., 2024).

We further examined two stricter variance thresholds, $\epsilon = 0.05$ (208 dimensions) and $\epsilon = 0.01$ (271 dimensions). Figure 4 shows that reconstructions are visually convincing at 176 dimensions and improve slightly with more dimensions, becoming nearly perfect at 271. We anticipate that incorporating more expressive transformations, such as rational quadratic spline flows (Durkan et al., 2019), could further refine the dimensionality estimation and improve reconstruction quality by better handling the optimization challenges associated with learning the distribution under the isometry and Hessian constraints.

Overall, these experiments illustrate that our approach scales to real-world data, provided we incorporate the Hessian vector product regularization term when using non-affine flows. These results mark significant progress toward robust data-

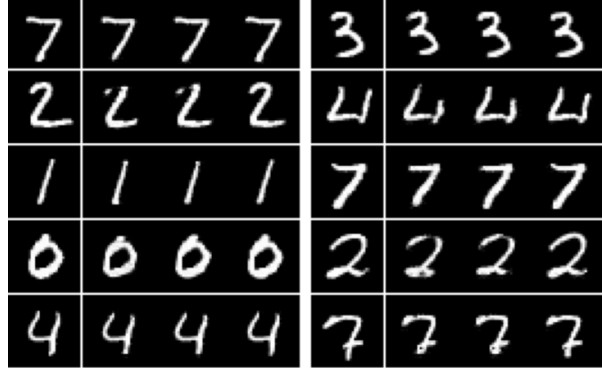

*Figure 4.* RAE reconstructions on MNIST. The leftmost column shows original images, while the next three show reconstructions for decreasing variance thresholds $\epsilon$ (0.1, 0.05, 0.01), using 176, 208, and 271 latent dimensions, respectively. Reconstructions are clear at 176 dimensions and nearly indistinguishable from the originals at 271.

driven Riemannian geometry in high-dimensional settings.

## 7. Conclusions

In this work we have taken a first step towards a practical data-driven Riemannian geometry framework, striking a balance between scalability of training a data-driven Riemannian structure and of evaluating its corresponding manifold mappings. We have considered a family of unimodal probability densities whose negative log-likelihoods are compositions of strongly convex functions and diffeomorphisms, and sought to learn them. We have shown that once these unimodal densities are learned, the proposed score-based pullback geometry provides closed-form geodesics that pass through the data support and an interpretable Riemannian autoencoder with error bounds that estimates the intrinsic dimension of the data. Finally, to learn the distribution we have proposed an adaptation to normalizing flow training. Through numerical experiments on *Euclidean* and *image* datasets, we have shown that these modifications are crucial for extracting geometric information, and that our framework not only generates high-quality geodesics across the data support, but also accurately estimates the intrinsic dimension of the approximate data manifold while constructing a global chart, even in high-dimensional settings.

Although the framework is theoretically best suited for unimodal distributions, it performs remarkably well on real multimodal distributions, albeit with a slight overestimation of intrinsic dimensionality. This highlights its practical utility for extracting geometry from real data. However, extending the formulation to better handle multimodal distributions remains an important direction for future work.

This work paves the way for scalable learning of data geome-

try, unlocking applications that were previously out of reach. It enables efficient computation of manifold maps such as geodesics and distances, with significant potential for advancing deep metric learning. Furthermore, it introduces Riemannian auto-encoders with interpretable latent spaces that effectively capture the intrinsic structure of data. Looking ahead, it paves the way for Riemannian optimization directly on the data manifold by enabling the computation of intrinsic gradients, with the potential to revolutionize inverse problem-solving and push the boundaries of controllable generative modeling.

## Acknowledgements

GB acknowledges support from the EPSRC Roseships grant. ZS acknowledges support from the Cantab Capital Institute for the Mathematics of Information, Christs College and the Trinity Henry Barlow Scholarship scheme. CBS acknowledges support from the Philip Leverhulme Prize, the Royal Society Wolfson Fellowship, the EPSRC advanced career fellowship EP/V029428/1, EPSRC Grants EP/S026045/1 and EP/T003553/1, EP/N014588/1, EP/T017961/1, the Wellcome Innovator Awards 215733/Z/19/Z and 221633/Z/20/Z, the European Union Horizon 2020 research and innovation programme under the Marie Skłodowska-Curie Grant agreement No. 777826 NoMADS, the Cantab Capital Institute for the Mathematics of Information and the Alan Turing Institute.

## Impact Statement

This paper presents work whose goal is to advance the field of Machine Learning and Riemannian Representation Learning. There are many potential societal consequences of our work, none which we feel must be specifically highlighted here.

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

# A. Generalizations and proofs of Proposition 3.1 and Theorem 3.3

Proposition 3.1 is a special case of the result below.

**Proposition A.1.** *Let $\varphi : \mathbb{R}^d \to \mathbb{R}^d$ be a smooth diffeomorphism and let $\psi : \mathbb{R}^d \to \mathbb{R}$ be a smooth strongly convex function, whose Fenchel conjugate is denoted by $\psi^\star : \mathbb{R}^d \to \mathbb{R}$. Next, consider the $\ell^2$-pullback manifolds $(\mathbb{R}^d, (\cdot, \cdot)^{\nabla\psi\circ\varphi})$ and $(\mathbb{R}^d, (\cdot, \cdot)^\varphi)$ defined through metric tensor fields*

$$(\Xi, \Phi)_{\mathbf{x}}^{\nabla\psi\circ\varphi} := (D_{\mathbf{x}}\nabla\psi \circ \varphi[\Xi], D_{\mathbf{x}}\nabla\psi \circ \varphi[\Phi])_2, \quad \text{and} \quad (\Xi, \Phi)_{\mathbf{x}}^\varphi := (D_{\mathbf{x}}\varphi[\Xi], D_{\mathbf{x}}\varphi[\Phi])_2. \tag{24}$$

*Then,*

(i) *the distance $d_{\mathbb{R}^d}^{\nabla\psi\circ\varphi} : \mathbb{R}^d \times \mathbb{R}^d \to \mathbb{R}$ on $(\mathbb{R}^d, (\cdot, \cdot)^{\nabla\psi\circ\varphi})$ is given by*

$$d_{\mathbb{R}^d}^{\nabla\psi\circ\varphi}(\mathbf{x}, \mathbf{y}) = \|(\nabla\psi \circ \varphi)(\mathbf{x}) - (\nabla\psi \circ \varphi)(\mathbf{y})\|_2. \tag{25}$$

*In addition, if $\psi$ is of the form Equation (4)*

$$d_{\mathbb{R}^d}^{\nabla\psi\circ\varphi}(\mathbf{x}, \mathbf{y}) = \|\varphi(\mathbf{x}) - \varphi(\mathbf{y})\|_{\mathbf{A}^{-2}} := \|\mathbf{A}^{-1}(\varphi(\mathbf{x}) - \varphi(\mathbf{y}))\|_2. \tag{26}$$

(ii) *length-minimising geodesics $\gamma_{\mathbf{x},\mathbf{y}}^{\nabla\psi\circ\varphi} : [0, 1] \to \mathbb{R}^d$ on $(\mathbb{R}^d, (\cdot, \cdot)^{\nabla\psi\circ\varphi})$ are given by*

$$\gamma_{\mathbf{x},\mathbf{y}}^{\nabla\psi\circ\varphi}(t) = (\varphi^{-1} \circ \nabla\psi^\star)((1 - t)(\nabla\psi \circ \varphi)(\mathbf{x}) + t(\nabla\psi \circ \varphi)(\mathbf{y})). \tag{27}$$

*In addition, if $\psi$ is of the form Equation (4)*

$$\gamma_{\mathbf{x},\mathbf{y}}^{\nabla\psi\circ\varphi}(t) = \gamma_{\mathbf{x},\mathbf{y}}^\varphi(t) = \varphi^{-1}((1 - t)\varphi(\mathbf{x}) + t\varphi(\mathbf{y})). \tag{28}$$

(iii) *the exponential map $\exp_{\mathbf{x}}^{\nabla\psi\circ\varphi}(\cdot) : \mathcal{T}_{\mathbf{x}}\mathbb{R}^d \to \mathbb{R}^d$ on $(\mathbb{R}^d, (\cdot, \cdot)^{\nabla\psi\circ\varphi})$ is given by*

$$\exp_{\mathbf{x}}^{\nabla\psi\circ\varphi}(\Xi_{\mathbf{x}}) = (\varphi^{-1} \circ \nabla\psi^\star)((\nabla\psi \circ \varphi)(\mathbf{x}) + D_{\varphi(\mathbf{x})}\nabla\psi[D_{\mathbf{x}}\varphi[\Xi_{\mathbf{x}}]]). \tag{29}$$

*In addition, if $\psi$ is of the form Equation (4)*

$$\exp_{\mathbf{x}}^{\nabla\psi\circ\varphi}(\Xi_{\mathbf{x}}) = \exp_{\mathbf{x}}^\varphi(\Xi_{\mathbf{x}}) = \varphi^{-1}(\varphi(\mathbf{x}) + D_{\mathbf{x}}\varphi[\Xi_{\mathbf{x}}]). \tag{30}$$

(iv) *the logarithmic map $\log_{\mathbf{x}}^{\nabla\psi\circ\varphi}(\cdot) : \mathbb{R}^d \to \mathcal{T}_{\mathbf{x}}\mathbb{R}^d$ on $(\mathbb{R}^d, (\cdot, \cdot)^{\nabla\psi\circ\varphi})$ is given by*

$$\log_{\mathbf{x}}^{\nabla\psi\circ\varphi}\mathbf{y} = D_{\varphi(\mathbf{x})}\varphi^{-1}[D_{(\nabla\psi\circ\varphi)(\mathbf{x})}\nabla\psi^\star[(\nabla\psi \circ \varphi)(\mathbf{y}) - (\nabla\psi \circ \varphi)(\mathbf{x})]]. \tag{31}$$

*In addition, if $\psi$ is of the form Equation (4)*

$$\log_{\mathbf{x}}^{\nabla\psi\circ\varphi}\mathbf{y} = \log_{\mathbf{x}}^\varphi\mathbf{y} = D_{\varphi(\mathbf{x})}\varphi^{-1}[\varphi(\mathbf{y}) - \varphi(\mathbf{x})]. \tag{32}$$

*Proof of Proposition 3.1.* First note that $\nabla\psi \circ \varphi$ is a diffeomorphism with inverse $\varphi^{-1} \circ \nabla\psi^\star$. Then, Equations (25), (27), (29) and (31) follow directly from (Diepeveen, 2024, Prop. 2.1).

Next, if $\psi$ is of the form Equation (4), i.e.,

$$\psi(\mathbf{x}) = \frac{1}{2}\mathbf{x}^\top \mathbf{A}^{-1}\mathbf{x},$$

we have that its Fenchel conjugate is given by

$$\psi^\star(\mathbf{y}) = \frac{1}{2}\mathbf{y}^\top \mathbf{A}\mathbf{y}. \tag{33}$$

So both $\nabla\psi(\mathbf{x}) = \mathbf{A}^{-1}\mathbf{x}$ and $\nabla\psi^\star(\mathbf{y}) = \mathbf{A}\mathbf{y}$ are linear mappings, from which follows that they cancel to identity everywhere and yield Equations (26), (28), (30) and (32). $\square$

Similarly, Theorem 3.3 is a special case of the result below.

**Theorem A.2.** *Let $\varphi : \mathbb{R}^d \to \mathbb{R}^d$ be a smooth diffeomorphism and let $\psi : \mathbb{R}^d \to \mathbb{R}$ be a smooth strongly convex function, whose Fenchel conjugate is denoted by $\psi^\star : \mathbb{R}^d \to \mathbb{R}$. Next, consider the function $f : \mathbb{R}^d \to \mathbb{R}^{d \times d}$ given by*

$$f(\mathbf{z}) := D_{\mathbf{z}} \nabla \psi^\star + \sum_{i=1}^{d} \mathbf{z}_i \partial_i D_{(\cdot)} \nabla \psi^\star. \tag{34}$$

*Finally, let $\mathbf{x}, \mathbf{y} \in \mathbb{R}^d$ be vectors and assume that for all vectors*

$$\mathbf{z} \in \{(1-t)(\nabla \psi \circ \varphi)(\mathbf{x}) + t(\nabla \psi \circ \varphi)(\mathbf{y}) \mid t \in [0,1]\} \subset \mathbb{R}^d$$

*the matrix $f(\mathbf{z})$ is positive definite.*

*Then, mapping*

$$t \mapsto \psi(\varphi(\gamma_{\mathbf{x},\mathbf{y}}^{\nabla \psi \circ \varphi}(t))), \quad t \in [0,1] \tag{35}$$

*is strongly convex, where $\gamma_{\mathbf{x},\mathbf{y}}^{\nabla \psi \circ \varphi}$ is the geodesic between $\mathbf{x}$ and $\mathbf{y}$ under the Riemannian structure $(\mathbb{R}^d, (\cdot, \cdot)^{\nabla \psi \circ \varphi})$.*

*In addition, if $\psi$ is of the form Equation (4) the mapping Equation (35) is strongly convex for any $\mathbf{x}, \mathbf{y} \in \mathbb{R}^d$.*

*Proof.* By Equation (27) in Proposition 3.1 we have

$$\psi(\varphi(\gamma_{\mathbf{x},\mathbf{y}}^{\nabla \psi \circ \varphi}(t))) = \psi(\varphi((\varphi^{-1} \circ \nabla \psi^\star)((1-t)(\nabla \psi \circ \varphi)(\mathbf{x}) + t(\nabla \psi \circ \varphi)(\mathbf{y}))))$$
$$= \psi(\nabla \psi^\star((1-t)(\nabla \psi \circ \varphi)(\mathbf{x}) + t(\nabla \psi \circ \varphi)(\mathbf{y}))). \tag{36}$$

So the claim holds if on the linear subspace

$$\{(1-t)(\nabla \psi \circ \varphi)(\mathbf{x}) + t(\nabla \psi \circ \varphi)(\mathbf{y}) \mid t \in [0,1]\} \subset \mathbb{R}^d \tag{37}$$

the function $\psi \circ \nabla \psi^\star$ is convex.

Next, note that the Hessian of $\psi \circ \nabla \psi^\star$ satisfies

$$D_{\mathbf{z}} \nabla(\psi \circ \nabla \psi^\star) = f(\mathbf{z}). \tag{38}$$

By assumption $f(\mathbf{z})$ is positive definite for all $\mathbf{z}$ in the subspace Equation (37). In other words, on this subspace $\psi(\nabla \psi^\star(\mathbf{z}))$ is positive definite, which implies strong convexity and yields the main claim.

The claim for the special case of $\psi$ is of the form Equation (4) follows directly, because

$$f(\mathbf{z}) = \mathbf{A}, \tag{39}$$

which is always positive definite.

$\square$

# B. Proof of Theorem 4.1

**Auxiliary lemma**

**Lemma B.1.** *Let $\varphi : \mathbb{R}^d \to \mathbb{R}^d$ be a smooth diffeomorphism and let $\psi : \mathbb{R}^d \to \mathbb{R}$ be a quadratic function of the form Equation (4) with diagonal $\mathbf{A} \in \mathbb{R}^{d \times d}$. Furthermore, let $p : \mathbb{R}^d \to \mathbb{R}$ be the corresponding probability density of the form Equation (2). Finally, consider $\varepsilon \in [0,1]$ and the mappings $E_\varepsilon : \mathbb{R}^d \to \mathbb{R}^{d_\varepsilon}$ and $D_\varepsilon : \mathbb{R}^{d_\varepsilon} \to \mathbb{R}^d$ in Equations (13) and (14) with $d_\varepsilon \in [d]$ as in Equation (12).*

*Then, for any $\alpha \in [0,1)$ and any $\beta \in [0, 1-\alpha)$*

$$\mathbb{E}_{\mathbf{X} \sim p}[d_{\mathbb{R}^d}^\varphi(D_\varepsilon(E_\varepsilon(\mathbf{X})), \mathbf{X})^2 e^{\frac{\alpha}{2} \varphi(\mathbf{X})^\top \mathbf{A}^{-1} \varphi(\mathbf{X})}] \leq \varepsilon \frac{C_{\beta,\varphi}^2 C_{\beta,\varphi}^3}{1-\alpha-\beta} \left( \frac{1+\beta}{1-\alpha-\beta} \right)^{\frac{d}{2}} \sum_{i=1}^{d} \mathbf{a}_i, \tag{40}$$

*where*

$$C^3_{\beta,\varphi} := \sup_{\mathbf{x} \in \mathbb{R}^d} \{|\det(D_{\varphi(\mathbf{x})}\varphi^{-1})|e^{-\frac{\beta}{2}\varphi(\mathbf{x})^\top \mathbf{A}^{-1}\varphi(\mathbf{x})}\}, \tag{41}$$

*and*

$$C^2_{\beta,\varphi} := \sup_{\mathbf{x} \in \mathbb{R}^d} \{|\det(D_{\mathbf{x}}\varphi)|e^{-\frac{\beta}{2}\varphi(\mathbf{x})^\top \mathbf{A}^{-1}\varphi(\mathbf{x})}\}. \tag{42}$$

*Proof.* We need to distinct two cases: (i) $d_\varepsilon = d$ and (ii) $1 \le d_\varepsilon < d$

(i) If $d_\varepsilon = d$ we have that $D_\varepsilon(E_\varepsilon(\mathbf{x})) = \mathbf{x}$ for any $\mathbf{x} \in \mathbb{R}^d$. In other words

$$\mathbb{E}_{\mathbf{X} \sim p}[d^\varphi_{\mathbb{R}^d}(D_\varepsilon(E_\varepsilon(\mathbf{X})), \mathbf{X})^2 e^{\frac{\alpha}{2}\varphi(\mathbf{X})^\top \mathbf{A}^{-1}\varphi(\mathbf{X})}] = 0 \le \varepsilon \frac{C^2_{\beta,\varphi}C^3_{\beta,\varphi}}{1-\alpha-\beta}\left(\frac{1+\beta}{1-\alpha-\beta}\right)^{\frac{d}{2}}\sum_{i=1}^d \mathbf{a}_i. \tag{43}$$

(ii) Next, we consider the case $1 \le d_\varepsilon < d$. First, notice that we can rewrite

$$\|\varphi(D_\varepsilon(E_\varepsilon(\mathbf{x}))) - \varphi(\mathbf{x})\|_2^2 \overset{\text{Equations (13) and (14)}}{=} \|\sum_{k=1}^{d_\varepsilon}(\varphi(\mathbf{x}), \mathbf{e}^{i_k})_2\mathbf{e}^{i_k} - \varphi(\mathbf{x})\|_2^2 = \|\sum_{k=d_\varepsilon+1}^{d}(\varphi(\mathbf{x}), \mathbf{e}^{i_k})_2\mathbf{e}^{i_k}\|_2^2$$

$$\overset{\text{orthogonality}}{=} \sum_{k=d_\varepsilon+1}^{d}\|(\varphi(\mathbf{x}), \mathbf{e}^{i_k})_2\mathbf{e}^{i_k}\|_2^2 = \sum_{k=d_\varepsilon+1}^{d}(\varphi(\mathbf{x}), \mathbf{e}^{i_k})_2^2 = \sum_{k=d_\varepsilon+1}^{d}\varphi(\mathbf{x})_{i_k}^2. \tag{44}$$

Moreover, we define

$$C := \int_{\mathbb{R}^d} e^{-\frac{1}{2}\varphi(\mathbf{x})^\top \mathbf{A}^{-1}\varphi(\mathbf{x})}\mathrm{d}\mathbf{x}. \tag{45}$$

Then,

$$\mathbb{E}_{\mathbf{X} \sim p}[d^\varphi_{\mathbb{R}^d}(D_\varepsilon(E_\varepsilon(\mathbf{X})), \mathbf{X})^2 e^{\frac{\alpha}{2}\varphi(\mathbf{X})^\top \mathbf{A}^{-1}\varphi(\mathbf{X})}] = \frac{\int_{\mathbb{R}^d}\|\varphi(D_\varepsilon(E_\varepsilon(\mathbf{x}))) - \varphi(\mathbf{x})\|_2^2 e^{-(\frac{1}{2}-\frac{\alpha}{2})\varphi(\mathbf{x})^\top \mathbf{A}^{-1}\varphi(\mathbf{x})}\mathrm{d}\mathbf{x}}{\int_{\mathbb{R}^d}e^{-\frac{1}{2}\varphi(\mathbf{x})^\top \mathbf{A}^{-1}\varphi(\mathbf{x})}\mathrm{d}\mathbf{x}}$$

$$\overset{\text{Equation (45)}}{=} \frac{1}{C}\int_{\mathbb{R}^d}\|\varphi(D_\varepsilon(E_\varepsilon(\mathbf{x}))) - \varphi(\mathbf{x})\|_2^2 e^{-(\frac{1}{2}-\frac{\alpha}{2})\varphi(\mathbf{x})^\top \mathbf{A}^{-1}\varphi(\mathbf{x})}\mathrm{d}\mathbf{x}$$

$$\overset{\text{Equation (44)}}{=} \frac{1}{C}\int_{\mathbb{R}^d}\sum_{k=d_\varepsilon+1}^{d}\varphi(\mathbf{x})_{i_k}^2 e^{-(\frac{1}{2}-\frac{\alpha}{2})\varphi(\mathbf{x})^\top \mathbf{A}^{-1}\varphi(\mathbf{x})}\mathrm{d}\mathbf{x} = \frac{1}{C}\sum_{k=d_\varepsilon+1}^{d}\int_{\mathbb{R}^d}\varphi(\mathbf{x})_{i_k}^2 e^{-(\frac{1}{2}-\frac{\alpha}{2})\varphi(\mathbf{x})^\top \mathbf{A}^{-1}\varphi(\mathbf{x})}\mathrm{d}\mathbf{x}$$

$$\overset{\mathbf{x}=\varphi^{-1}(\mathbf{y})}{=} \frac{1}{C}\sum_{k=d_\varepsilon+1}^{d}\int_{\mathbb{R}^d}\mathbf{y}_{i_k}^2 e^{-(\frac{1}{2}-\frac{\alpha}{2})\mathbf{y}^\top \mathbf{A}^{-1}\mathbf{y}}|\det(D_{\mathbf{y}}\varphi^{-1})|\mathrm{d}\mathbf{y}$$

$$= \frac{1}{C}\sum_{k=d_\varepsilon+1}^{d}\int_{\mathbb{R}^d}\mathbf{y}_{i_k}^2 e^{-(\frac{1}{2}-\frac{\alpha}{2}-\frac{\beta}{2})\mathbf{y}^\top \mathbf{A}^{-1}\mathbf{y}}|\det(D_{\mathbf{y}}\varphi^{-1})|e^{-\frac{\beta}{2}\mathbf{y}^\top \mathbf{A}^{-1}\mathbf{y}}\mathrm{d}\mathbf{y}$$

$$\le \frac{\sup_{\mathbf{y} \in \mathbb{R}^d}\{|\det(D_{\mathbf{y}}\varphi^{-1})|e^{-\frac{\beta}{2}\mathbf{y}^\top \mathbf{A}^{-1}\mathbf{y}}\}}{C}\sum_{k=d_\varepsilon+1}^{d}\int_{\mathbb{R}^d}\mathbf{y}_{i_k}^2 e^{-(\frac{1}{2}-\frac{\alpha}{2}-\frac{\beta}{2})\mathbf{y}^\top \mathbf{A}^{-1}\mathbf{y}}\mathrm{d}\mathbf{y}$$

$$\overset{\text{Equation (41)}}{=} \frac{C^2_{\beta,\varphi}}{C}\sum_{k=d_\varepsilon+1}^{d}\int_{\mathbb{R}^d}\mathbf{y}_{i_k}^2 e^{-(\frac{1}{2}-\frac{\alpha}{2}-\frac{\beta}{2})\mathbf{y}^\top \mathbf{A}^{-1}\mathbf{y}}\mathrm{d}\mathbf{y} = \frac{C^2_{\beta,\varphi}}{C}\sum_{k=d_\varepsilon+1}^{d}\int_{\mathbb{R}^d}\mathbf{y}_{i_k}^2 e^{-(\frac{1}{2}-\frac{\alpha}{2}-\frac{\beta}{2})\sum_{j=1}^d \frac{\mathbf{y}_j^2}{\mathbf{a}_j}}\mathrm{d}\mathbf{y}$$

$$= \frac{C^2_{\beta,\varphi}}{C}\sum_{k=d_\varepsilon+1}^{d}\int_{\mathbb{R}}\mathbf{y}_{i_k}^2 e^{-(\frac{1}{2}-\frac{\alpha}{2}-\frac{\beta}{2})\frac{\mathbf{y}^2}{\mathbf{a}_{i_k}}}\mathrm{d}\mathbf{y}_{i_k}\int_{\mathbb{R}^{d-1}}e^{-(\frac{1}{2}-\frac{\alpha}{2}-\frac{\beta}{2})\sum_{j\ne i_k}^d \frac{\mathbf{y}_j^2}{\mathbf{a}_j}}\mathrm{d}\mathbf{y}_1\ldots\mathrm{d}\mathbf{y}_{i_k-1}\mathrm{d}\mathbf{y}_{i_k+1}\ldots\mathrm{d}\mathbf{y}_d$$

$$= \frac{C^2_{\beta,\varphi}}{C}\sum_{k=d_\varepsilon+1}^{d}\frac{\mathbf{a}_{i_k}}{(1-\alpha-\beta)}\int_{\mathbb{R}}e^{-(\frac{1}{2}-\frac{\alpha}{2}-\frac{\beta}{2})\frac{\mathbf{y}^2}{\mathbf{a}_{i_k}}}\mathrm{d}\mathbf{y}_{i_k}\int_{\mathbb{R}^{d-1}}e^{-(\frac{1}{2}-\frac{\alpha}{2}-\frac{\beta}{2})\sum_{j\ne i_k}^d \frac{\mathbf{y}_j^2}{\mathbf{a}_j}}\mathrm{d}\mathbf{y}_1\ldots\mathrm{d}\mathbf{y}_{i_k-1}\mathrm{d}\mathbf{y}_{i_k+1}\ldots\mathrm{d}\mathbf{y}_d$$

$$= \frac{C_{\beta,\varphi}^2}{C} \sum_{k=d_\varepsilon+1}^{d} \frac{\mathbf{a}_{i_k}}{(1-\alpha-\beta)} \int_{\mathbb{R}^d} e^{-(\frac{1}{2}-\frac{\alpha}{2}-\frac{\beta}{2})\mathbf{y}^\top \mathbf{A}^{-1}\mathbf{y}} \mathrm{d}\mathbf{y}$$

$$= \frac{C_{\beta,\varphi}^2}{C} \sum_{k=d_\varepsilon+1}^{d} \frac{\mathbf{a}_{i_k}}{(1-\alpha-\beta)} \Big(\frac{1+\beta}{1-\alpha-\beta}\Big)^{\frac{d}{2}} \int_{\mathbb{R}^d} e^{-(\frac{1}{2}+\frac{\beta}{2})\mathbf{y}^\top \mathbf{A}^{-1}\mathbf{y}} \mathrm{d}\mathbf{y}$$

$$\overset{\mathbf{y}=\varphi(\mathbf{x})}{=} \frac{C_{\beta,\varphi}^2}{C} \sum_{k=d_\varepsilon+1}^{d} \frac{\mathbf{a}_{i_k}}{(1-\alpha-\beta)} \Big(\frac{1+\beta}{1-\alpha-\beta}\Big)^{\frac{d}{2}} \int_{\mathbb{R}^d} e^{-(\frac{1}{2}+\frac{\beta}{2})\varphi(\mathbf{x})^\top \mathbf{A}^{-1}\varphi(\mathbf{x})} |\det(D_\mathbf{x}\varphi)| \mathrm{d}\mathbf{x}$$

$$= \frac{C_{\beta,\varphi}^2}{C} \sum_{k=d_\varepsilon+1}^{d} \frac{\mathbf{a}_{i_k}}{(1-\alpha-\beta)} \Big(\frac{1+\beta}{1-\alpha-\beta}\Big)^{\frac{d}{2}} \int_{\mathbb{R}^d} e^{-\frac{1}{2}\varphi(\mathbf{x})^\top \mathbf{A}^{-1}\varphi(\mathbf{x})} |\det(D_\mathbf{x}\varphi)| e^{-\frac{\beta}{2}\varphi(\mathbf{x})^\top \mathbf{A}^{-1}\varphi(\mathbf{x})} \mathrm{d}\mathbf{x}$$

$$\leq \frac{C_{\beta,\varphi}^2 \sup_{\mathbf{x}\in\mathbb{R}^d}\{|\det(D_\mathbf{x}\varphi)| e^{-\frac{\beta}{2}\varphi(\mathbf{x})^\top \mathbf{A}^{-1}\varphi(\mathbf{x})}\}}{C} \sum_{k=d_\varepsilon+1}^{d} \frac{\mathbf{a}_{i_k}}{(1-\alpha-\beta)} \Big(\frac{1+\beta}{1-\alpha-\beta}\Big)^{\frac{d}{2}} \int_{\mathbb{R}^d} e^{-\frac{1}{2}\varphi(\mathbf{x})^\top \mathbf{A}^{-1}\varphi(\mathbf{x})} \mathrm{d}\mathbf{x}$$

$$\overset{\text{Equation (42)}}{=} \frac{C_{\beta,\varphi}^2 C_{\beta,\varphi}^3}{C} \sum_{k=d_\varepsilon+1}^{d} \frac{\mathbf{a}_{i_k}}{(1-\alpha-\beta)} \Big(\frac{1+\beta}{1-\alpha-\beta}\Big)^{\frac{d}{2}} \int_{\mathbb{R}^d} e^{-\frac{1}{2}\varphi(\mathbf{x})^\top \mathbf{A}^{-1}\varphi(\mathbf{x})} \mathrm{d}\mathbf{x}$$

$$\overset{\text{Equation (45)}}{=} \frac{C_{\beta,\varphi}^2 C_{\beta,\varphi}^3}{1-\alpha-\beta} \Big(\frac{1+\beta}{1-\alpha-\beta}\Big)^{\frac{d}{2}} \sum_{k=d_\varepsilon+1}^{d} \mathbf{a}_{i_k}$$

$$\overset{\text{Equation (12)}}{\leq} \varepsilon \frac{C_{\beta,\varphi}^2 C_{\beta,\varphi}^3}{1-\alpha-\beta} \Big(\frac{1+\beta}{1-\alpha-\beta}\Big)^{\frac{d}{2}} \sum_{i=1}^{d} \mathbf{a}_i. \quad (46)$$

$$\square$$

**Proof of the theorem**

*Proof of Theorem 4.1.* First, consider the Taylor approximation

$$\varphi^{-1}(\varphi(\mathbf{y})) - \varphi^{-1}(\varphi(\mathbf{y})) = D_{\varphi(\mathbf{x})}\varphi^{-1}[\varphi(\mathbf{y}) - \varphi(\mathbf{x})] + \mathcal{O}(\|\varphi(\mathbf{y}) - \varphi(\mathbf{x})\|_2^2)$$
$$= D_{\varphi(\mathbf{x})}\varphi^{-1}[\varphi(\mathbf{y}) - \varphi(\mathbf{x})] + \mathcal{O}(d_{\mathbb{R}^d}^\varphi(\mathbf{y},\mathbf{x})^2). \quad (47)$$

Moreover, we define

$$C := \int_{\mathbb{R}^d} e^{-\frac{1}{2}\varphi(\mathbf{x})^\top \mathbf{A}^{-1}\varphi(\mathbf{x})} \mathrm{d}\mathbf{x}. \quad (48)$$

Subsequently, notice that

$$\mathbb{E}_{\mathbf{X}\sim p}[\|D_{\varphi(\mathbf{X})}\varphi^{-1}[\varphi(D_\varepsilon(E_\varepsilon(\mathbf{X}))) - \varphi(\mathbf{X})]\|_2^2]$$

$$= \frac{1}{C} \int_{\mathbb{R}^d} \|D_{\varphi(\mathbf{x})}\varphi^{-1}[\varphi(D_\varepsilon(E_\varepsilon(\mathbf{x}))) - \varphi(\mathbf{x})]\|_2^2 e^{-\frac{1}{2}\varphi(\mathbf{x})^\top \mathbf{A}^{-1}\varphi(\mathbf{x})} \mathrm{d}\mathbf{x}$$

$$\leq \frac{1}{C} \int_{\mathbb{R}^d} \|D_{\varphi(\mathbf{x})}\varphi^{-1}\|_2^2 \|\varphi(D_\varepsilon(E_\varepsilon(\mathbf{x}))) - \varphi(\mathbf{x})\|_2^2 e^{-\frac{1}{2}\varphi(\mathbf{x})^\top \mathbf{A}^{-1}\varphi(\mathbf{x})} \mathrm{d}\mathbf{x}$$

$$\leq \frac{\sup_{\mathbf{x}\in\mathbb{R}^d}\{\|D_{\varphi(\mathbf{x})}\varphi^{-1}\|_2^2 e^{-\frac{\beta}{2}\varphi(\mathbf{x})^\top \mathbf{A}^{-1}\varphi(\mathbf{x})}\}}{C} \int_{\mathbb{R}^d} \|\varphi(D_\varepsilon(E_\varepsilon(\mathbf{x}))) - \varphi(\mathbf{x})\|_2^2 e^{-(\frac{1}{2}-\frac{\beta}{2})\varphi(\mathbf{x})^\top \mathbf{A}^{-1}\varphi(\mathbf{x})} \mathrm{d}\mathbf{x}$$

$$\overset{\text{Equation (17)}}{=} \frac{C_{\beta,\varphi}^1}{C} \int_{\mathbb{R}^d} \|\varphi(D_\varepsilon(E_\varepsilon(\mathbf{x}))) - \varphi(\mathbf{x})\|_2^2 e^{\frac{\beta}{2}\varphi(\mathbf{x})^\top \mathbf{A}^{-1}\varphi(\mathbf{x})} e^{-\frac{1}{2}\varphi(\mathbf{x})^\top \mathbf{A}^{-1}\varphi(\mathbf{x})} \mathrm{d}\mathbf{x}$$

$$= C_{\beta,\varphi}^1 \mathbb{E}_{\mathbf{X}\sim p}[d_{\mathbb{R}^d}^\varphi(D_\varepsilon(E_\varepsilon(\mathbf{X})), \mathbf{X})^2 e^{\frac{\beta}{2}\varphi(\mathbf{X})^\top \mathbf{A}^{-1}\varphi(\mathbf{X})}]$$

$$\overset{\text{Lemma B.1}}{\leq} \varepsilon \frac{C_{\beta,\varphi}^1 C_{\beta,\varphi}^2 C_{\beta,\varphi}^3}{1-2\beta} \Big(\frac{1+\beta}{1-2\beta}\Big)^{\frac{d}{2}} \sum_{i=1}^{d} \mathbf{a}_i. \quad (49)$$

Then,

$$
\begin{aligned}
\mathbb{E}_{\mathbf{X}\sim p}[\|D_\varepsilon(E_\varepsilon(\mathbf{X})) - \mathbf{X}\|_2^2] &= \mathbb{E}_{\mathbf{X}\sim p}[\|\varphi^{-1}(\varphi(D_\varepsilon(E_\varepsilon(\mathbf{X})))) - \varphi^{-1}(\varphi(\mathbf{X}))\|_2^2] \\
&\overset{Equation\ (47)}{=} \mathbb{E}_{\mathbf{X}\sim p}[\|D_{\varphi(\mathbf{X})}\varphi^{-1}[\varphi(D_\varepsilon(E_\varepsilon(\mathbf{X}))) - \varphi(\mathbf{X})] + \mathcal{O}(d_{\mathbb{R}^d}^\varphi(D_\varepsilon(E_\varepsilon(\mathbf{X})), \mathbf{X})^2)\|_2^2] \\
&= \mathbb{E}_{\mathbf{X}\sim p}[\|D_{\varphi(\mathbf{X})}\varphi^{-1}[\varphi(D_\varepsilon(E_\varepsilon(\mathbf{X}))) - \varphi(\mathbf{X})]\|_2^2 + \mathcal{O}(d_{\mathbb{R}^d}^\varphi(D_\varepsilon(E_\varepsilon(\mathbf{X})), \mathbf{X})^3)] \\
&\overset{Equation\ (49)}{\leq} \varepsilon\frac{C_{\beta,\varphi}^1 C_{\beta,\varphi}^2 C_{\beta,\varphi}^3}{1-2\beta}\left(\frac{1+\beta}{1-2\beta}\right)^{\frac{d}{2}}\sum_{i=1}^d \mathbf{a}_i + o(\varepsilon), \quad (50)
\end{aligned}
$$

which yields the claim as $\beta$ was arbitrary. $\qquad\square$

## C. Training Details

This section describes the configuration parameters necessary to reproduce our experiments. The experiments are categorized into two groups: *Euclidean* datasets (Sinusoid and Hemisphere) and *image* datasets (Gaussian Blobs and MNIST). All experiments share some common parameters, listed below, while dataset-specific parameters are provided in Table 2.

**Common Parameters:**

- **Optimizer:** Adam with `betas` = (0.9, 0.99), `eps` = $1 \times 10^{-8}$, and weight decay of $1 \times 10^{-5}$.

- **Gradient Clipping:** Gradient norm clipped to 1.0.

- **Model Architecture:**

  - **Euclidean Datasets:** A normalizing flow composed of affine coupling layers, where each layer transforms part of the input while leaving the remaining dimensions unchanged. The parameters of the affine transformations are parameterized by a ResNet with 64 hidden features and 2 residual blocks. Transformations alternate across dimensions at each step.
  - **Image Datasets:** A normalizing flow composed of three *levels*, each beginning with a squeeze operation that redistributes spatial dimensions into channels. Within each level, we apply a stack of seven flow steps, leading to a total of 21 flow steps across all levels. Each flow step consists of:
    * An ActNorm layer.
    * A $1 \times 1$ invertible convolution for channel mixing.
    * An affine coupling transform, where the parameters of the affine transformation are parameterized by a convolutional ResNet. This ResNet has 96 hidden channels and 3 residual blocks.

*Table 2.* Training configurations for each experiment.

| Dataset | Flow Steps | Epochs | Batch Size | $\lambda_{\text{iso}}$ | $\lambda_{\text{vol}}$ | $\lambda_{\text{hess}}$ | Learning Rate |
|---|---|---|---|---|---|---|---|
| Sinusoid(1,3) | 8 | 1000 | 64 | 1.0 | 1.0 | - | $3 \times 10^{-4}$ |
| Sinusoid(2,3) | 8 | 1000 | 64 | 1.0 | 1.0 | - | $3 \times 10^{-4}$ |
| Sinusoid(5,20) | 24 | 2000 | 128 | 1.2 | 2.5 | - | $4 \times 10^{-4}$ |
| Hemisphere(2,3) | 8 | 2000 | 64 | 1.0 | 1.0 | - | $4 \times 10^{-4}$ |
| Hemisphere(5,20) | 12 | 2000 | 64 | 0.75 | 1.2 | - | $4 \times 10^{-4}$ |
| 20-Blobs | 21 | 500 | 64 | 100 | 1.0 | 0.5 | $2 \times 10^{-4}$ |
| 50-Blobs | 21 | 500 | 64 | 100 | 1.0 | 0.5 | $2 \times 10^{-4}$ |
| 100-Blobs | 21 | 500 | 64 | 100 | 1.0 | 0.5 | $2 \times 10^{-4}$ |
| MNIST | 21 | 500 | 64 | 100 | 2.0 | 1.0 | $2 \times 10^{-4}$ |

# D. Additional Experimental Results

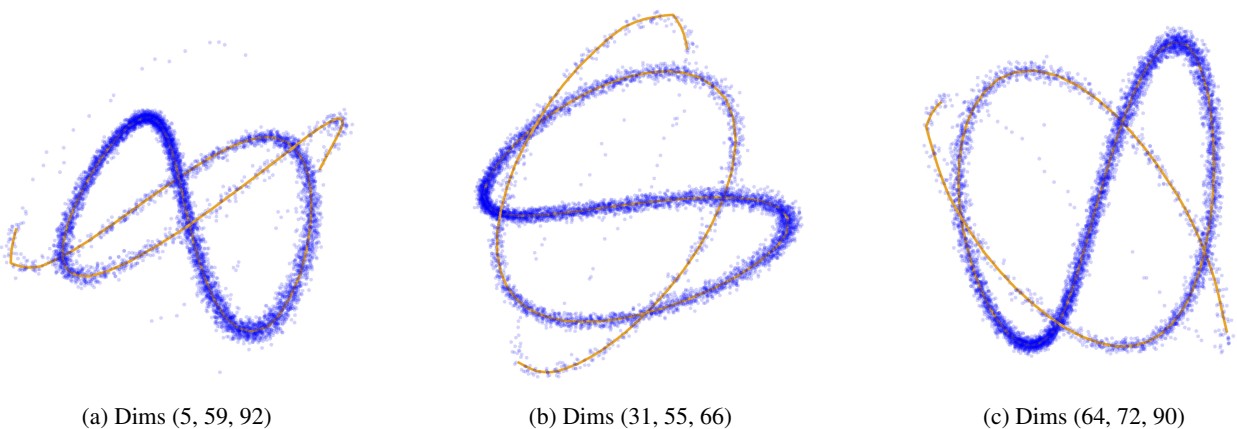

(a) Dims (5, 59, 92)

(b) Dims (31, 55, 66)

(c) Dims (64, 72, 90)

*Figure 5.* Approximate data manifold learned by the Riemannian autoencoder for the Sinusoid(1, 100) dataset. The orange curves depict the learned manifold, while blue points represent training data.

**Hemisphere (5,20)**

(a) Learned variances in decreasing order.

(b) Reconstruction error for three latent orders.

**Sinusoid (5,20)**

(c) Learned variances in decreasing order.

(d) Reconstruction error for three latent orders.

*Figure 6.* Learned variances and reconstruction errors for Hemisphere(5,20) and Sinusoid(5,20). Left: variances in decreasing order. Right: average $\ell^2$ reconstruction error vs. latent dimensions. Errors are shown for three variance-based orders: **blue** (decreasing variance), **green** (increasing variance), and **red** (random).

## E. Error Metrics for Evaluation of Pullback Geometries

**Geodesic Error.** The geodesic error measures the difference between geodesics on the learned and ground truth pullback manifolds. Given two points $\mathbf{x}_0, \mathbf{x}_1 \in \mathbb{R}^d$, let $\gamma^{\varphi_{\theta_2}}_{\mathbf{x}_0, \mathbf{x}_1}(t)$ and $\gamma^{\varphi_{\mathrm{GT}}}_{\mathbf{x}_0, \mathbf{x}_1}(t)$ denote the geodesics induced by the learned map $\varphi_{\theta_2}$ and the ground truth map $\varphi_{\mathrm{GT}}$, respectively, where $t \in [0, 1]$.

The geodesic error is calculated as the mean Euclidean distance between the learned and ground truth geodesics over $N$ pairs of points:

$$\text{Geodesic Error} = \frac{1}{N}\sum_{i=1}^{N}\frac{1}{T}\sum_{k=1}^{T}\left\| \gamma^{\varphi_{\theta_2}}_{\mathbf{x}_0^{(i)}, \mathbf{x}_1^{(i)}}(t_k) - \gamma^{\varphi_{\mathrm{GT}}}_{\mathbf{x}_0^{(i)}, \mathbf{x}_1^{(i)}}(t_k) \right\|_2,$$

where $T$ is the number of time steps used to discretize the geodesic, and $t_k = \frac{k-1}{T-1}$ for $k = 1, \dots, T$.

This metric captures the average discrepancy between the learned and ground truth geodesics, reflecting the accuracy of the learned pullback manifold.

**Variation Error.** The variation error quantifies the sensitivity of the geodesic computation under small perturbations to one of the endpoints. For two points $\mathbf{x}_0, \mathbf{x}_1 \in \mathbb{R}^d$, let $\mathbf{z} = \mathbf{x}_1 + \Delta\mathbf{x}$, where $\Delta\mathbf{x}$ is a random variable sampled from the Gaussian distribution:

$$\Delta\mathbf{x} \sim \mathcal{N}(\mathbf{0}, 0.1^2\mathbf{I}),$$

with mean $\mathbf{0}$ and covariance $0.1^2\mathbf{I}$, where $\mathbf{I}$ is the identity matrix. Define $\gamma^{\varphi_{\theta_2}}_{\mathbf{x}_0, \mathbf{x}_1}(t)$ and $\gamma^{\varphi_{\theta_2}}_{\mathbf{x}_0, \mathbf{z}}(t)$ as the geodesics from $\mathbf{x}_0$ to $\mathbf{x}_1$ and $\mathbf{z}$, respectively, induced by the learned map $\varphi_{\theta_2}$.

The variation error is calculated as the mean Euclidean distance between the geodesic from $\mathbf{x}_0$ to $\mathbf{x}_1$ and the perturbed geodesic from $\mathbf{x}_0$ to $\mathbf{z}$:

$$\text{Variation Error} = \frac{1}{N}\sum_{i=1}^{N}\frac{1}{T}\sum_{k=1}^{T}\left\| \gamma^{\varphi_{\theta_2}}_{\mathbf{x}_0^{(i)}, \mathbf{x}_1^{(i)}}(t_k) - \gamma^{\varphi_{\theta_2}}_{\mathbf{x}_0^{(i)}, \mathbf{z}^{(i)}}(t_k) \right\|_2,$$

where $N$ is the number of sampled point pairs, $T$ is the number of time steps used to discretize the geodesic, and $t_k = \frac{k-1}{T-1}$ for $k = 1, \dots, T$.

This metric evaluates the robustness of the learned geodesic against small perturbations, providing insight into the stability of the learned manifold.

## F. Dataset Construction Details

In this section, we provide a detailed explanation of the construction of the datasets used in our experiments. We organize the datasets into two categories based on the experimental sections in which they are used.

### F.1. Datasets for Manifold Mapping Experiments

In our manifold mapping experiments (Section 6.1), we use the following datasets illustrated in Figure 7:

- *Single Banana Dataset*: A two-dimensional dataset shaped like a curved banana.

- *Squeezed Single Banana Dataset*: A variant of the Single Banana with a tighter bend.

- *River Dataset*: A more complex 2D dataset resembling the meandering path of a river.

Each dataset is constructed by defining specific diffeomorphisms $\varphi$ and convex quadratic functions $\psi$, then sampling from the resulting probability density using Langevin Monte Carlo Markov Chain (MCMC) with Metropolis-Hastings correction. The probability density function is defined as:

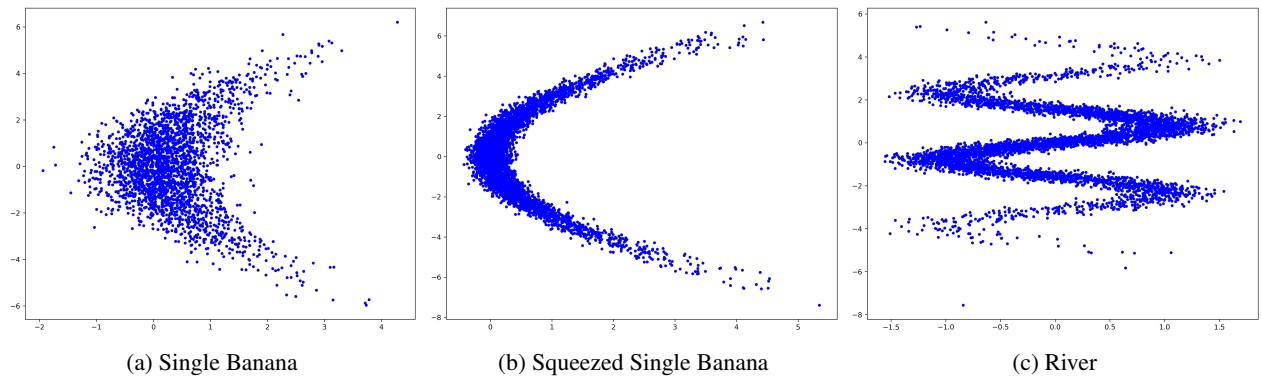

(a) Single Banana      (b) Squeezed Single Banana      (c) River

*Figure 7.* Visualization of the datasets used in our manifold mapping experiments.

$$p(\mathbf{x}) \propto e^{-\psi(\varphi(\mathbf{x}))}, \tag{51}$$

where the strongly convex function $\psi$ is given by:

$$\psi(\mathbf{v}) = \frac{1}{2}\mathbf{v}^\top \mathbf{A}^{-1}\mathbf{v}, \tag{52}$$

and $A$ is a positive-definite diagonal matrix. The specific choices of $\varphi$ and $A$ for each dataset determine its geometric properties.

### F.1.1. DIFFEOMORPHISMS AND CONVEX QUADRATIC FUNCTIONS

The key differences between the datasets arise from the diffeomorphism $\varphi$ and the covariance matrix $\mathbf{A}$ used in the sampling process. Below, we describe the specific settings for each dataset.

**1. Single Banana Dataset**

- Diffeomorphism:

$$\varphi(\mathbf{x}) = \begin{pmatrix} x_1 - ax_2^2 - z \\ x_2 \end{pmatrix}$$

where $a = \frac{1}{9}$ and $z = 0$.

- Covariance matrix:

$$\mathbf{A} = \begin{pmatrix} \frac{1}{4} & 0 \\ 0 & 4 \end{pmatrix}$$

**2. Squeezed Single Banana Dataset**

- Diffeomorphism: Same as the Single Banana Dataset.

- Covariance matrix:

$$\mathbf{A} = \begin{pmatrix} \frac{1}{81} & 0 \\ 0 & 4 \end{pmatrix}$$

**3. River Dataset**

- Diffeomorphism:

$$\varphi(\mathbf{x}) = \begin{pmatrix} x_1 - \sin(ax_2) - z \\ x_2 \end{pmatrix}$$

where $a = 2$ and $z = 0$.

- Covariance matrix:

$$\mathbf{A} = \begin{pmatrix} \frac{1}{25} & 0 \\ 0 & 3 \end{pmatrix}$$

### F.1.2. DATASET GENERATION ALGORITHM

Algorithm 1 outlines the dataset generation process for all three datasets. The specific diffeomorphisms and quadratic functions differ for each dataset.

---

**Algorithm 1** General Dataset Generation Algorithm

---

**Require:** Number of samples $N$, MCMC steps $T$, Step size $\delta$, Diffeomorphism $\varphi$, Covariance matrix $\mathbf{\Lambda}$
**Ensure:** Dataset $\{\mathbf{x}_1, \mathbf{x}_2, \ldots, \mathbf{x}_N\}$
 1: Initialize: Set initial state $\mathbf{x}_0 = \mathbf{0} \in \mathbb{R}^2$.
 2: **for** $i = 1$ to $N$ **do**
 3:     $\mathbf{x} = \mathbf{x}_0$
 4:     **for** $k = 1$ to $T$ **do**
 5:         Compute the score function $\nabla_{\mathbf{x}} \log p_{\text{target}}(\mathbf{x})$.
 6:         Propose $\mathbf{x}' = \mathbf{x} + \frac{\delta^2}{2} \nabla_{\mathbf{x}} \log p_{\text{target}}(\mathbf{x}) + \delta\boldsymbol{\eta}$, where $\boldsymbol{\eta} \sim \mathcal{N}(\mathbf{0}, \mathbf{I}_2)$.
 7:         Compute the forward kernel:
$$K_{\text{forward}} = \frac{|\mathbf{x} - \mathbf{x}' + \frac{\delta^2}{2}\nabla_{\mathbf{x}'} \log p_{\text{target}}(\mathbf{x}')|^2}{2\delta^2}$$
 8:         Compute the reverse kernel:
$$K_{\text{reverse}} = \frac{|\mathbf{x}' - \mathbf{x} + \frac{\delta^2}{2}\nabla_{\mathbf{x}} \log p_{\text{target}}(\mathbf{x})|^2}{2\delta^2}$$
 9:         Compute the Metropolis-Hastings acceptance probability:
$$A = \min\left(1, \frac{p_{\text{target}}(\mathbf{x}')}{p_{\text{target}}(\mathbf{x})} \exp\left(-K_{\text{forward}} + K_{\text{reverse}}\right)\right)$$
10:         Accept $\mathbf{x}'$ with probability $A$; else set $\mathbf{x}' = \mathbf{x}$.
11:         Update $\mathbf{x} = \mathbf{x}'$.
12:     **end for**
13:     Store the final $\mathbf{x}$ as sample $\mathbf{x}_i$.
14: **end for**

---

### F.2. Datasets for Riemannian Autoencoder Experiments

In the Riemannian autoencoder experiments (Section 6.2), we use the following datasets:

- *Hemisphere(d', d)* Dataset: Samples drawn from the upper hemisphere of a $d'$-dimensional unit sphere and embedded into $\mathbb{R}^d$ via a random isometric mapping.

- *Sinusoid(d', d)* Dataset: Generated by applying sinusoidal transformations to $d'$-dimensional latent variables, resulting in a complex, nonlinear manifold in $\mathbb{R}^d$.

### F.3. Hemisphere($d'$, $d$) Dataset

The *Hemisphere($d'$, $d$)* dataset consists of samples drawn from the upper hemisphere of a $d'$-dimensional unit sphere, which are then embedded into a $d$-dimensional ambient space using a random isometric embedding. Below are the steps involved in constructing this dataset.

**1. Sampling from the Upper Hemisphere**  We begin by sampling points from the upper hemisphere of the $d'$-dimensional unit sphere $S_+^{d'} \subset \mathbb{R}^{d'+1}$. The upper hemisphere is defined as:

$$S_+^{d'} = \left\{\mathbf{x} \in \mathbb{R}^{d'+1} : \|\mathbf{x}\| = 1,\ x_1 \geq 0\right\}.$$

The first angular coordinate $\theta_1$ is sampled from a Beta distribution with shape parameters $\alpha = 5$ and $\beta = 5$, scaled to the interval $\left[0, \frac{\pi}{2}\right]$. This sampling method emphasizes points near the "equator" of the hemisphere. The remaining angular coordinates $\theta_2, \ldots, \theta_{d'}$ are sampled uniformly from the interval $[0, \pi]$:

$$\theta_1 \sim \text{Beta}(5,5) \cdot \left(\frac{\pi}{2}\right), \quad \theta_i \sim \text{Uniform}(0, \pi), \text{ for } i = 2, \ldots, d'.$$

**2. Conversion to Cartesian Coordinates** Next, each sampled point in spherical coordinates is converted into Cartesian coordinates in $\mathbb{R}^{d'+1}$ using the following transformation equations:

$$x_1 = \cos(\theta_1), \quad x_2 = \sin(\theta_1)\cos(\theta_2), \quad \ldots, \quad x_{d'+1} = \sin(\theta_1)\sin(\theta_2)\cdots\sin(\theta_{d'}).$$

This conversion ensures that the sampled points lie on the surface of the unit sphere in $(d'+1)$-dimensional space.

**3. Random Isometric Embedding into $\mathbb{R}^d$** After sampling points on the hemisphere in $\mathbb{R}^{d'+1}$, the points are embedded into a $d$-dimensional ambient space ($d \geq d' + 1$) using a random isometric embedding. The embedding process is as follows:

1. Generate a random matrix $\mathbf{A} \in \mathbb{R}^{d \times (d'+1)}$, where each entry is sampled from a standard normal distribution $\mathcal{N}(0, 1)$.

2. Perform a QR decomposition on matrix $\mathbf{A}$ to obtain $\mathbf{Q} \in \mathbb{R}^{d \times (d'+1)}$:

$$\mathbf{A} = \mathbf{Q}\mathbf{R}.$$

The columns of $\mathbf{Q}$ form an orthonormal basis for a $(d'+1)$-dimensional subspace of $\mathbb{R}^d$, ensuring that $\mathbf{Q}$ defines an isometric embedding from $\mathbb{R}^{d'+1}$ into $\mathbb{R}^d$. This guarantees that distances and angles are preserved during the mapping, maintaining the geometric structure of the original space within the higher-dimensional ambient space.

3. Use matrix $\mathbf{Q}$ to map each sample $\mathbf{x} \in \mathbb{R}^{d'+1}$ into the ambient space:

$$\mathbf{y} = \mathbf{Q}\mathbf{x},$$

where $\mathbf{y} \in \mathbb{R}^d$ are the embedded samples.

---

**Algorithm 2** Hemisphere($d'$, $d$) Dataset Generation

---

1: **Input:** Intrinsic dimension $d'$, ambient dimension $d$, number of samples $n$, Beta distribution parameters $\alpha = 5$, $\beta = 5$
2: **Output:** Dataset $\mathbf{Y} \in \mathbb{R}^{n \times d}$
3: **Step 1: Generate Random Isometric Embedding**
4: Generate a random matrix $\mathbf{A} \in \mathbb{R}^{d \times (d'+1)}$ with entries from $\mathcal{N}(0, 1)$
5: Perform QR decomposition on $\mathbf{A}$ to obtain $\mathbf{Q} \in \mathbb{R}^{d \times (d'+1)}$:

$$\mathbf{A} = \mathbf{QR}$$

6: **Step 2: Construct Dataset**
7: **for** $i = 1$ to $n$ **do**
8:     **Step 2.1: Sample Spherical Coordinates**
9:     Sample the first angular coordinate $\theta_1$ from a scaled Beta distribution:

$$\theta_1 \sim \text{Beta}(\alpha, \beta) \cdot \left( \frac{\pi}{2} \right)$$

10:     Sample the remaining angular coordinates $\theta_2, \ldots, \theta_{d'}$ from a uniform distribution:

$$\theta_i \sim \text{Uniform}(0, \pi), \quad \text{for } i = 2, \ldots, d'$$

11:     **Step 2.2: Convert to Cartesian Coordinates**
12:     Convert the spherical coordinates to Cartesian coordinates $\mathbf{x}_i \in \mathbb{R}^{d'+1}$ using:

$$x_1 = \cos(\theta_1), \quad x_2 = \sin(\theta_1)\cos(\theta_2), \ldots, \quad x_{d'+1} = \sin(\theta_1)\sin(\theta_2)\cdots\sin(\theta_{d'}).$$

13:     **Step 2.3: Embed Sample $\mathbf{x}_i$ into Ambient Space**
14:     Map the sample $\mathbf{x}_i$ to the ambient space using:
$$\mathbf{y}_i = \mathbf{Q}\mathbf{x}_i$$

15:     Append $\mathbf{y}_i$ to the dataset $\mathbf{Y}$
16: **end for**
17: **Return:** The final dataset $\mathbf{Y} = [\mathbf{y}_1, \mathbf{y}_2, \ldots, \mathbf{y}_n]$

---

### F.4. Sinusoid($d'$, $d$) Dataset

The *Sinusoid*($d'$, $d$) dataset represents a $d'$-dimensional manifold embedded in $d$-dimensional space through nonlinear sinusoidal transformations. Below are the detailed steps involved in constructing this dataset.

**1. Sampling Latent Variables**   The latent variables $\mathbf{z} \in \mathbb{R}^{d'}$ are sampled from a multivariate Gaussian distribution with zero mean and isotropic variance, as follows:
$$\mathbf{z} \sim \mathcal{N}\left(0, \sigma_m^2 I_{d'}\right),$$
where $\sigma_m^2$ controls the variance along each intrinsic dimension, and $I_{d'}$ is the $d' \times d'$ identity matrix. The value of $\sigma_m^2$ is set to 3 for our experiments.

**2. Defining Ambient Coordinates with Sinusoidal Transformations**   For each of the $d - d'$ ambient dimensions, we construct a shear vector $\mathbf{a}_j \in \mathbb{R}^{d'}$, with its elements drawn uniformly from the interval $[1, 2]$:

$$\mathbf{a}_j \sim \text{Uniform}(1, 2)^{d'}, \quad \text{for } j = 1, \ldots, d - d'.$$

The shear vectors $\mathbf{a}_j$ apply a fixed linear transformation to the latent space $\mathbf{z} \in \mathbb{R}^{d'}$, determining how the latent variables influence each ambient dimension. These vectors, sampled once for each of the $d - d'$ ambient dimensions, modulate the scale and periodicity of the sinusoidal transformation.

Each ambient coordinate $x_j$ is generated as a sinusoidal function of the inner product between $\mathbf{a}_j$ and $\mathbf{z}$, with a small Gaussian noise added for regularization.

$$x_j = \sin\left(\mathbf{a}_j^\top \mathbf{z}\right) + \epsilon_j,$$

where $\epsilon_j \sim \mathcal{N}(0, \sigma_a^2)$ is Gaussian noise with variance $\sigma_a^2$. In our experiments, we set $\sigma_a^2 = 10^{-3}$.

**3. Constructing the Dataset Samples**   The final samples $\mathbf{y} \in \mathbb{R}^d$ are formed by concatenating the ambient coordinates $x_1, x_2, \ldots, x_{d-d'}$ with the latent variables $z_1, z_2, \ldots, z_{d'}$:

$$\mathbf{y} = [x_1, x_2, \ldots, x_{d-d'}, z_1, z_2, \ldots, z_{d'}]^\top.$$

---

**Algorithm 3** Sinusoid($d'$, $d$) Dataset Generation

---

1: **Input:** Intrinsic dimension $d'$, ambient dimension $d$, number of samples $n$, variance $\sigma_m^2 = 3$, noise variance $\sigma_a^2 = 10^{-3}$
2: **Output:** Dataset $\mathbf{Y} \in \mathbb{R}^{n \times d}$
3: **Step 1: Generate Shear Vectors**
4: **for** $j = 1$ to $d - d'$ **do**
5:     Sample shear vector $\mathbf{a}_j \in \mathbb{R}^{d'}$ from Uniform$(1, 2)^{d'}$
6: **end for**
7: **Step 2: Construct Dataset**
8: **for** $i = 1$ to $n$ **do**
9:     **Step 2.1: Sample Latent Variables**
10:     Generate latent variables $\mathbf{z}_i \in \mathbb{R}^{d'}$ from a multivariate Gaussian:

$$\mathbf{z}_i \sim \mathcal{N}(0, \sigma_m^2 \cdot I_{d'})$$

11:     **Step 2.2: Compute Ambient Coordinates for Sample** $i$
12:     **for** $j = 1$ to $d - d'$ **do**
13:         Compute ambient coordinate $x_j$ for the $i$-th sample:

$$x_j = \sin\left(\mathbf{a}_j^\top \mathbf{z}_i\right) + \epsilon_j, \quad \epsilon_j \sim \mathcal{N}(0, \sigma_a^2)$$

14:     **end for**
15:     **Step 2.3: Form Final Sample** $\mathbf{y}_i$
16:     Concatenate the ambient coordinates $\mathbf{x} = [x_1, x_2, \ldots, x_{d-d'}]$ and the latent variables $\mathbf{z}_i$ to form the final sample $\mathbf{y}_i \in \mathbb{R}^d$:

$$\mathbf{y}_i = [x_1, x_2, \ldots, x_{d-d'}, z_1, z_2, \ldots, z_{d'}]^\top$$

17:     Append $\mathbf{y}_i$ to the dataset $\mathbf{Y}$
18: **end for**
19: **Return:** The final dataset $\mathbf{Y} = [\mathbf{y}_1, \mathbf{y}_2, \ldots, \mathbf{y}_n]$

---

### F.5. Gaussian Blobs Image Manifold

The *Gaussian Blobs Image Manifold* dataset defines an image manifold with a controllable intrinsic dimension $d'$ (number of Gaussian blobs) embedded in a 1024-dimensional ambient space ($32 \times 32$ pixel images). The dataset is constructed as follows:

1. $d'$ Gaussian blob centers are randomly selected from a $32 \times 32$ grid without replacement. These centers are fixed and remain the same for all points in the dataset.

2. For each image, a Gaussian mixture is generated by centering Gaussian distributions at the fixed locations, with standard deviations sampled uniformly from a specified range $[s_{\min}, s_{\max}]$.

3. The normalized density of the Gaussian mixture defines the intensity values of the 2D image. Each combination of standard deviations uniquely defines a point on the manifold.

---

**Algorithm 4** Gaussian Blobs Image Manifold Dataset Generation

---

**Require:** Number of samples $N$, intrinsic dimension $d'$, image size $I$, standard deviation range $[s_{\min}, s_{\max}]$, random seed $S$

**Ensure:** Dataset $\{\mathbf{x}_1, \mathbf{x}_2, \ldots, \mathbf{x}_N\}$ with $\mathbf{x}_i \in \mathbb{R}^{I \times I}$

1: Initialize: Set random seed $S$ for reproducibility.
2: Generate $d'$ Gaussian centers by sampling without replacement from the $I \times I$ pixel grid. **These centers are fixed for all samples in the dataset.**
3: **for** $i = 1$ to $N$ **do**
4:     Initialize an empty image $\mathbf{x}_i$ of size $I \times I$.
5:     **for** $j = 1$ to $d'$ **do**
6:         Select the $j$-th Gaussian center $(x_j, y_j)$ from the preselected centers.
7:         Sample standard deviation $s_j$ uniformly from $[s_{\min}, s_{\max}]$.
8:         Compute the Gaussian density over the entire image grid:

$$g(x, y) = \frac{1}{\sqrt{2\pi}s_j} \exp\left(-\frac{(x - x_j)^2 + (y - y_j)^2}{2s_j^2}\right)$$

9:         Add the Gaussian density to the image: $\mathbf{x}_i(x, y) \mathrel{+}= g(x, y)$.
10:     **end for**
11:     Normalize $\mathbf{x}_i$ to the range $[0, 1]$:

$$\mathbf{x}_i = \frac{\mathbf{x}_i - \min(\mathbf{x}_i)}{\max(\mathbf{x}_i) - \min(\mathbf{x}_i)}$$

12:     Store the normalized image $\mathbf{x}_i$ as a sample.
13: **end for**

---

# G. Data Manifold Approximation

The learned manifold, shown in orange in Figure 1, is the set $D_\epsilon(\mathcal{U})$, where $D_\epsilon$ is the RAE decoder Equation (14), the set $\mathcal{U}$ in the latent space is the open set given by

$$\mathcal{U} = \prod_{i=1}^{d_\epsilon}(-3\sqrt{\mathbf{a}_{u_i}}, 3\sqrt{\mathbf{a}_{u_i}})$$

and $\mathbf{a}_{u_1}, \ldots, \mathbf{a}_{u_{d_\epsilon}}$ are the $d_\epsilon$ highest learned variances corresponding to the ones used in the RAE construction.

To visualize this in practice, we construct a mesh grid by linearly sampling each latent dimension from $-3\sqrt{\mathbf{a}_{u_i}}$ to $+3\sqrt{\mathbf{a}_{u_i}}$, for $i = 1, \ldots, d_\epsilon$, where $d_\epsilon$ is the number of significant latent dimensions. Practically, the off-manifold latent dimensions (those corresponding to negligible variances) are set to zero. The decoder $D_\epsilon$ then maps this grid from $\mathcal{U}$ back to $\mathbb{R}^d$, generating an approximation of the data manifold, as illustrated in Figure 1.

# H. Explanation for the Need of Higher Regularity in Non-Affine Flows

As described in Section 6.2.2, introducing a small regularization term involving the Hessian vector product of the flow can substantially improve the RAE's ability to assign higher variances to the on-manifold (semantically meaningful) directions and suppress variances in off-manifold directions for *non-affine* flows. The core issue arises from the fact that, for *non-affine* normalizing flows, the second-order derivatives of $\varphi_{\theta_2}$ appear in the Hessian of the modeled log-density in a way that can adversely impact the performance of the RAE if left unregularized, as we shall see below.

## H.1. Hessian of the Log-Density and Intrinsic Dimensionality

Recently, Stanczuk et al. (2024) formally proved—and Stanczuk et al. (2024); Kadkhodaie et al. (2024); Wenliang & Moran (2022) independently demonstrated empirically—that when data concentrate locally around a lower-dimensional manifold of dimension $k$, the *Jacobian* of the score function or equivalently, the *Hessian* of the log-density has $k$ vanishing singular values. The eigenvectors corresponding to these vanishing singular values span the local tangent space of the data manifold. Consequently, for a trained model, large singular values of $\nabla_\mathbf{x}^2 \log p_{\theta_1, \theta_2}(\mathbf{x})$ indicate *off-manifold* directions, while near-zero

singular values reveal *on-manifold* directions—those spanning the locally flat subspace in which the data reside.

## H.2. Analyzing the Hessian of the Log-Density of Our Model

With the background knowledge on the connection between the Hessian of the log-density and the local intrinsic structure in mind, let's investigate the Hessian of the logdensity of our model under the assumption that $\varphi_{\theta_2}$ is local isometry and hence volume preserving as well. The modeled density function is

$$p_{\theta_1,\theta_2}(\mathbf{x}) \; = \; p_{\theta_1}\big(\varphi_{\theta_2}(\mathbf{x})\big) \; \Big|\det\!\big(D_{\mathbf{x}}\varphi_{\theta_2}(\mathbf{x})\big)\Big|, \tag{53}$$

where $D_{\mathbf{x}}\varphi_{\theta_2}(\mathbf{x})$ is the *Jacobian* of $\varphi_{\theta_2}$ at $\mathbf{x}$ and $p_{\theta_1}\big(\varphi_{\theta_2}(\mathbf{x})\big)= \mathcal{N}(\mathbf{x}; \mathbf{0}, \mathbf{A}_{\theta_1})$. Since $\varphi_{\theta_2}$ is volume preserving, the absolute value of the Jacobian determinant is $1$, simplifying the density to:

$$p_{\theta_1,\theta_2}(\mathbf{x}) \; = \; p_{\theta_1}\big(\varphi_{\theta_2}(\mathbf{x})\big), \tag{54}$$

Taking logarithms and differentiating yields the expression for the Hessian of the logdensity:

$$\nabla_{\mathbf{x}}^2 \log p_{\theta_1,\theta_2}(\mathbf{x}) \; = \; -\nabla_{\mathbf{x}}^2\varphi_{\theta_2}(\mathbf{x})\, \mathbf{A}_{\theta_1}^{-1}\, \varphi_{\theta_2}(\mathbf{x}) \; - \; \big(D_{\mathbf{x}}\varphi_{\theta_2}(\mathbf{x})\big)^{\top} \mathbf{A}_{\theta_1}^{-1} \big(D_{\mathbf{x}}\varphi_{\theta_2}(\mathbf{x})\big), \tag{55}$$

where $\nabla_{\mathbf{x}}^2\varphi_{\theta_2}(\mathbf{x})$ is the *Hessian* of $\varphi_{\theta_2}$. We see that the Hessian of the log-density consists of the sum of two terms:

• A Hessian vector product term:

$$- \nabla_{\mathbf{x}}^2\varphi_{\theta_2}(\mathbf{x})\, \mathbf{A}_{\theta_1}^{-1}\, \varphi_{\theta_2}(\mathbf{x}),$$

• A term that depends on the Jacobian and the learned covariance matrix $\mathbf{A}_{\theta_1}$:

$$- \big(D_{\mathbf{x}}\varphi_{\theta_2}(\mathbf{x})\big)^{\top} \mathbf{A}_{\theta_1}^{-1} \big(D_{\mathbf{x}}\varphi_{\theta_2}(\mathbf{x})\big).$$

When $\varphi_{\theta_2}$ is *affine* (e.g., composed of affine coupling layers only), the Hessian vector product term in equation 55 vanishes. Hence, the Hessian of the logdensity simplifies to:

$$\nabla_{\mathbf{x}}^2 \log p_{\theta_1,\theta_2}(\mathbf{x}) \; = \; -\big(D_{\mathbf{x}}\varphi_{\theta_2}(\mathbf{x})\big)^{\top} \mathbf{A}_{\theta_1}^{-1} \big(D_{\mathbf{x}}\varphi_{\theta_2}(\mathbf{x})\big), \tag{56}$$

Noting that the Jacobian of $\varphi_{\theta_2}$, denoted as $D_{\mathbf{x}}\varphi_{\theta_2}(\mathbf{x})$, is an orthogonal matrix due to $\varphi_{\theta_2}$ being a local isometry, and considering that $\mathbf{A}_{\theta_1}$ is a diagonal matrix with strictly positive entries, it becomes clear that the right-hand side of Equation equation 56 represents the eigendecomposition of the Hessian of the log-density. The magnitude of the eigenvalues of the Hessian directly influences how variance is allocated in the latent space. Large eigenvalues correspond to off-manifolds directions and therefore will be encoded by latent dimensions of small learned variance, while small eigenvalues correspond to on-manifold directions and will be encoded by latent dimensions of high learned variance. This analysis provides a clear and intuitive explanation of why the Riemannian Autoencoder effectively detects and encodes important semantic information in the latent dimensions associated with high learned variances when trained with an affine normalizing flow regularized for local isometry.

However, for *non-affine* flows—such as those incorporating $1 \times 1$ invertible convolutions after each affine coupling layer or rational quadratic splines—the Hessian vector product term can become significant and may *interfere with* the Jacobian term. This disruption can distort the learned manifold geometry, leading to an incorrect allocation of variances in the latent space. Specifically, the model may assign large variances to an increased number of latent dimensions and thus overestimate the intrinsic dimension. To address this, we add a Hessian vector product penalty to the loss, minimizing its contribution to the Hessian of the log-density and allowing the Riemannian Autoencoder to accurately capture the data manifold's lower-dimensional structure in $\mathbf{A}_{\theta_1}$.

Note that $\varphi_{\theta_2}$ maps $\mathbb{R}^d \to \mathbb{R}^d$, making its Hessian $\nabla_{\mathbf{x}}^2\varphi_{\theta_2}(\mathbf{x})$ a rank-3 tensor of shape $d \times d \times d$. Multiplying this Hessian by $\mathbf{A}_{\theta_1}^{-1} \varphi_{\theta_2}(\mathbf{x})$ results in a $d \times d$ matrix, whose computation becomes prohibitively expensive in high dimensions, even with optimized methods. To address this in our MNIST experiments, we randomly selected a dimension at each iteration,

computed the corresponding column of the Hessian vector product, and penalized its Euclidean norm (refer to Appendix I for additional details). This efficient regularization was enough to keep the Hessian term orders of magnitude smaller in Frobenius norm compared to the Jacobian term, allowing the Riemannian Autoencoder to perform effectively with *non-affine* flows.

## I. Computational Complexity of the Regularization Terms

In this section, we discuss the computational complexity and memory requirements for each of the additional regularization terms introduced in our objective. For simplicity, we denote the normalizing flow $\varphi_{\theta_2}(x)$ as $\phi(x)$ throughout this section:

$$
\begin{aligned}
\mathcal{L}_{\text{reg}}(\theta_1, \theta_2) = &\underbrace{\lambda_{\text{vol}}\, \mathbb{E}_{x \sim p_{\text{data}}}\left[\left(\log|\det(D_x\phi)|\right)^2\right]}_{\text{Volume Regularization}} \\
&+ \underbrace{\lambda_{\text{iso}}\, \mathbb{E}_{x \sim p_{\text{data}}}\left[\left\|(D_x\phi)^\top D_x\phi - \mathbf{I}_d\right\|_F^2\right]}_{\text{Isometry Regularization}} \\
&+ \underbrace{\lambda_{\text{hess}}\, \mathbb{E}_{x \sim p_{\text{data}}}\left[\left\|D_x^2\phi \cdot \Sigma_{\theta_1}^{-1}\phi(x)\right\|_2\right]}_{\text{Hessian Regularization}}.
\end{aligned}
$$

The **Hessian regularization** term is only applied when $\phi$ is a *non-affine* flow. For affine flows, the volume and isometry regularization terms are sufficient, as the second derivatives vanish in this case, making the Hessian penalty unnecessary.

The presented regularization objective is computationally feasible only for **low-dimensional** data, as evaluating the full Jacobian and the Hessian-vector product is expensive in both computation and memory for high dimensions. To address this, we present efficient approximations designed to significantly reduce computational and memory costs while preserving the effectiveness of the regularization. The details of these approximations, including their implementation and impact on scalability, are provided in this section.

### I.1. Volume Regularization

The volume regularization term

$$
\mathbb{E}_{x \sim p_{\text{data}}}\left[\left(\log|\det(D_x\phi)|\right)^2\right]
$$

relies on the log-determinant of the Jacobian, which is typically *already computed* during the forward pass in normalizing flows. Most flow architectures are designed so that $\log|\det(D_x\phi)|$ is tractable and inexpensive to obtain (e.g., via coupling transforms or autoregressive transforms).

Hence, **no extra gradient backprop** or additional memory overhead is needed for this volume penalty—it essentially comes for free from the standard normalizing flow likelihood computation.

### I.2. Isometry Regularization

The isometry regularization term,

$$
\mathbb{E}_{x \sim p_{\text{data}}}\left[\left\|(D_x\phi)^\top D_x\phi - \mathbf{I}_d\right\|_F^2\right],
$$

penalizes deviations of the Jacobian $D_x\phi$ from orthogonality. By enforcing this condition, the regularization ensures that the mapping $\phi$ approximates a local isometry, preserving distances and angles in the vicinity of each point.

**Full Objective.** The full implementation involves computing the $d \times d$ Jacobian matrix for each sample, which is both computationally and memory-intensive in high-dimensional settings. These constraints render the full objective impractical for high-dimensional data, limiting its application to low-dimensional scenarios where such costs remain manageable.

**Sliced Objective.** To address the scalability limitations of the full objective, we introduce the *sliced objective*. Instead of computing the full Jacobian, we approximate it using Jacobian-vector products (JVPs) with a small number $m$ of randomly

sampled orthonormal vectors. This approach significantly reduces computational and memory requirements while retaining the effectiveness of the regularization. The procedure is outlined in Algorithm 5.

---

**Algorithm 5** Sliced Isometry Regularization Objective

---

1: **Input:** Mapping $\phi$, batch of inputs $x \in \mathbb{R}^{B \times d}$, number of orthonormal vectors $m$, and device.
2: **Output:** Approximate orthogonality regularization loss $\mathcal{L}_{\text{iso}}$.
3: Generate a random matrix $R \in \mathbb{R}^{B \times d \times m}$ with entries sampled from $\mathcal{N}(0, 1)$.
4: Perform QR decomposition on $R$ for each batch sample to obtain orthonormal vectors $Q \in \mathbb{R}^{B \times d \times m}$.
5: Compute Jacobian-vector products (JVPs) for all $m$ orthonormal vectors $v_i \in Q$:

$$Jv_i = \nabla \phi(x) v_i \quad \text{for } i = 1, \ldots, m.$$

6: Construct the Gram matrix:

$$G[i, j] = (Jv_i)^\top (Jv_j) \quad \forall i, j \in \{1, \ldots, m\}.$$

7: Compute the deviation of $G$ from the identity matrix:

$$G - \mathbf{I}_m,$$

where $\mathbf{I}_m$ is the identity matrix of size $m \times m$.
8: Compute the Frobenius norm of the deviation for each sample:

$$\|G - \mathbf{I}_m\|_F^2.$$

9: Compute the batch mean of the Frobenius norms:

$$\mathcal{L}_{\text{iso}} = \frac{1}{B} \sum_{i=1}^{B} \|G_i - \mathbf{I}_m\|_F^2.$$

---

In practice, we employ PyTorch's efficient Jacobian-vector product implementation to compute $Jv_i$ efficiently. To further enhance performance, we leverage `torch.vmap` to vectorize computations over the batch dimension, enabling parallelized and streamlined evaluation of the regularization term across all samples in the batch. These optimizations substantially improve scalability, making the sliced objective well-suited for larger batch sizes and high-dimensional data.

Notably, our experiments demonstrate that using only $m = 2$ slicing orthonormal vectors is sufficient for efficient and effective isometry regularization. This approach effectively reduces the computational cost of the regularization term to 2 JVPs per sample, compared to $d$ full backpropagation passes required for the full objective. Consequently, the sliced objective enables the method to scale effectively to high-dimensional datasets.

### I.3. Hessian-Vector Product (HVP) Regularization

The Hessian-vector product regularization $\mathbb{E}_{x \sim p_{\text{data}}} \left[ \left\| D_x^2 \phi \cdot \Sigma_{\theta_1}^{-1} \phi(x) \right\|_2 \right]$, limits the influence of second-order derivatives on the Hessian of the log-density of the model distribution, enhancing the Riemannian Auto-encoder's ability to capture the intrinsic geometry of the data manifold with expressive non-affine flows. It is unnecessary for affine flows, where these terms naturally vanish. See Appendix H for details.

**Full Objective.** For each output dimension $j$ of $\phi$, the Hessian $D_x^2 \phi_j(x)$ is a $d \times d$ matrix, and the full Hessian across all $d$ outputs forms a rank-3 tensor of shape $\mathbb{R}^{d \times d \times d}$. As a result, even with optimized Hessian-vector product implementations, computing this term becomes infeasible for high-dimensional data.

**Sliced Objective.** To reduce computational overhead, we approximate the Hessian penalty by sampling a single dimension $j \in \{1, \ldots, d\}$ at each training iteration. Instead of minimizing the Frobenius norm of the full Hessian-vector product matrix, we compute and minimize the norm of a single column (of size $d \times 1$) corresponding to the sampled dimension $j$. This approach is lightweight and empirically sufficient for effective regularization.

---

**Algorithm 6** Sliced Hessian Regularization Objective

---

1: **Input:** Mapping $\phi$, batch of inputs $x \in \mathbb{R}^{B \times d}$, inverse covariance $\Sigma_{\theta_1}^{-1}$.
2: **Output:** Approximate Hessian regularization loss $\mathcal{L}_{\text{hess}}$.
3: Randomly sample a single output dimension $j \in \{1, \ldots, d\}$.
4: Compute the Hessian-vector product (HVP) for each batch sample:

$$\text{HVP}_j = D_x^2 \phi_j(x) \cdot \left(\Sigma_{\theta_1}^{-1} \phi(x)\right),$$

   using `torch.autograd.functional.hvp` or equivalent.
5: Compute the norm of the resulting $d \times 1$ vector for each batch sample:

$$\left\|\text{HVP}_j\right\|_2^2.$$

6: Compute the batch mean of the norms:

$$\mathcal{L}_{\text{hess}} = \frac{1}{B} \sum_{i=1}^{B} \left\|\text{HVP}_j[i]\right\|_2^2.$$

---

In our experiments, evaluating a single random column of the Hessian-vector product matrix per iteration and penalizing its norm was sufficient to maintain low Frobenius norm of the full $d \times d$ matrix, effectively regularizing the flow. The computation of the sliced Hessian-vector product in optimized deep learning libraries is equivalent to two backpropagation passes: one for the gradient and one for the vector-Jacobian product. Therefore, the sliced objective enables the method to scale effectively to complex, high-dimensional datasets.

