# OpenReview forum: "Score-based Pullback Riemannian Geometry: Extracting the Data Manifold Geometry using Anisotropic Flows"
_ICML.cc/2025/Conference — ICML 2025 poster_

### Official Review · Reviewer_M1sQ · 2025-03-09

**Overall Recommendation:** 4

**Summary:**

The manifold assumption states that data often reside in low-dimensional submanifolds of the original ambient space. The primary goal of this work is to learn the structure of this low-dimensional manifold and identify the intrinsic dimensionality. This is solved by considering unimodal densities obtained by deforming Gaussians under smooth diffeomorphisms and inferring the geometry of the sub-manifold given by regions of high likelihoods. By fitting this density on data in a manner akin to normalizing flows, one is able to learn the approximate geometry (e.g. Riemannian metric, geodesics, Levi-Civita connection, etc...) of this underlying manifold by pulling back with respect to the diffeomorphsim. Furthermore, since the learned density is modeled as the pullback of a Gaussian under this diffeomorphism, one can also apply PCA on the deformed space (where the data manifold is Euclidean) to infer the principal components and, hence, the intrinsic dimensionality of the data manifold.

## Update after rebuttal
The authors have convincingly addressed my original concerns regarding scalability benefits and the necessity of including anisotropy and isometry regularization in recovering the intrinsic dimensions. Hence, I have increased my evaluation of the paper to a 4 from my original score of 3.

**Claims And Evidence:**

From the theoretical side, the main claims made in this paper are:
- Closed form expressions of the Riemannian structure induced by the distribution $p(\mathbf{x}) \propto e^{-\psi(\varphi(\mathbf{x})))}$.
- Claim that the corresponding geodesics pass through high likelihood regions of $p(\mathbf{x})$ by demonstrating geodesic convexity of $\psi$.
- Error bound for Riemannian PCA.

The above claims are supported by rigorous proofs.

From the numerical side, the authors show that:
- The proposed model and learning algorithm successfully learns the geometry of the data manifold.
- The Riemannian PCA is able to learn low-dimensional representations of high-dimensional data.

For the first point, the authors only demonstrate this on benchmarks given by deformations of a unimodal Gaussian - an idealised setting considered in this model. They consider an ablation with regards to various components of the model, in particular, anisotropy and near-isometry. They demonstrate clearly the benefits of adding both components to achieve successful learning of the underlying geometry.

The second experiment is performed on various high dimensional datasets, where it is demonstrated that in the unimodal case, the Riemannian PCA successfully learns the intrinsic dimension. In the multimodal case (MNIST), which is not a natural setting for the proposed model, the algorithm is demonstrated to still work well, despite overestimating the intrinsic dimensions.

However, comparisons to other manifold learning methods are missing, which makes the claim that the proposed model _successfully balances scalability of manifold mapping and training cost_ (which seems to be the main claim that the paper makes) less evident.

**Essential References Not Discussed:**

I believe the authors cover the literature sufficiently well to motivate their work, however, I am not familiar enough with the literature in generative manifold learning to point out specific related works.

**Experimental Designs Or Analyses:**

I have checked the soundness of the experimental setup used in the paper, which is given in details in the supplementary. I do not see any obvious issues.

**Methods And Evaluation Criteria:**

The data and metrics used to assess the model are chosen appropriately.

**Other Comments Or Suggestions:**

I believe the term "score-based pullback Riemannian geometry" is used quite loosely here, as the model does not use the score function -  it is only related in the case when $\varphi$ is an isometry. I believe this can be misleading as readers might come in expecting the use of the score function obtained e.g. using score-based diffusion models, to build a Riemannian geometry that represents the data. However, this is not really what is done in this paper.

**Other Strengths And Weaknesses:**

__Strengths:__
- The paper provides a rigorous error analysis of the Riemannian PCA, which illuminates how the error shrinks with the $\epsilon$ parameter, which controls the number of principal components to use.
- Experimental results and figures show clear evidence that the model is capable of learning the Riemannian structure of the underlying data manifold, as well as the intrinsic dimension.

__Weaknesses:__
- Ultimately, I don't believe the authors adequately support their claim about _striking a good balance between scalability of training and evaluating manifold mappings_. While it makes sense that by considering a simplified model, this should be true, a more rigorous statement about how the proposed method is scalable would be useful. Additionally, it would be useful to see comparisons with the previous "unscalable" approaches in their experiments as baselines.
- It would also be useful to see if there are any limitations posed by restricting to unimodal distributions of the form $p(\mathbf{x}) \propto e^{-\psi(\varphi(\mathbf{x})))}$. Are there multimodal examples where the model fails to work? Or is it fairly robust, as shown in the MNIST example?
- To state simply, the proposed model is a normalizing flow with regularization. Hence, the inclusion of this regularization seems to be the key algorithmic innovation. While the first experiment shows the benefit of this regularization in learning the manifold mapping, it is not demonstrated in the second experiment for learning the intrinsic dimension of the data. It would be useful to understand if the results in the experiments in Section 6.2 are not just due to normalizing flows but if the added regularization is contributing to the observed positive results.

**Questions For Authors:**

1. Can you demonstrate the benefits in terms of scalability of the proposed method compared to previous works like Sorrenson et al. (2024), Diepeveen (2024), etc? I believe this is important in order to adequately support the claims in this paper.
2. How do the results in Section 6.2 compare when just using a normalising flow (NF) to obtain the pullback geometry? Having an ablation with NF, Anisotropic NF, Isometric NF would also be useful here to find out if the regularisations are complementing the NF to help discover the intrinsic dimension of the underlying Riemannian geometry.
3. What is the reconstruction error in Figure 3? Is it the MSE?

**Relation To Broader Scientific Literature:**

The present work is extension of ongoing work in data-driven Riemannian geometry, where the goal is to learn intrinsic low-dimensional manifold representations of data. Previous works have utilised generative models to learn the underlying manifold structure, however have limited applicability, due to difficulties in working with the learned geometry (as in Sorrenson et al. 2024) and/or difficulties in training (Diepeveen, 2004, etc...). The work seeks to find a model/algorithm that is scalable both in training time and deployment. However, in doing so, they limit their models to be coming from probability distributions of the form $p(\mathbf{x}) \propto e^{-\psi(\varphi(\mathbf{x})))}$ for quadratic $\psi$, which is essentially a family of probability distributions obtained by nonlinear deformations of the Gaussian (i.e. normalising flows).

I would however remark that the term "score-based" in the title can be a little misleading, since, as the authors also demonstrate, the proposed model is only equivalent to the pullback geometry with respect to the score function if and only if $\varphi$ is a Euclidean isometry. In the end, the training follows that of normalising flows with minor adaptations to accommodate approximate isometry.

**Theoretical Claims:**

I have checked the correctness of the closed form expressions of the induced Riemannian structure and geodesic convexity of $\psi$. I have not checked the proof details of the error analysis of Riemannian PCA (Appendix B).

---

> ### Author Rebuttal · Authors · 2025-03-31
>
> ### **Weaknesses**
>
> **1.)** Refer to our response to Q1.
>
> **2.)** While our parametric assumption indeed focuses on unimodal distributions of the form $p(\mathbf{x}) \propto e^{-\psi(\varphi(x))}$, we observed encouraging robustness in multimodal settings, as demonstrated by the MNIST dataset. Although MNIST was the only multimodal dataset tested, we believe it clearly illustrates the method's effectiveness in multimodal scenarios. A thorough investigation into our model's performance across a wider variety of multimodal datasets is an interesting direction we reserve for future work.
>
> **3.)** Refer to our response to Q2.
>
> ---
>
> ### **Questions for Authors**
>
> **Q1.)** **Sorrenson et al.** first fit a normalizing flow to the data to construct a density-weighted KNN graph. Dijkstra’s algorithm is applied to estimate initial geodesic paths, which are then refined by numerically solving the geodesic ODE using a separately trained score model. However, this pipeline does not scale beyond low-dimensional toy datasets (~25D), due to key bottlenecks: (i) the graph-based initialization becomes unreliable in higher dimensions (Section 5.2.1), and (ii) solving the geodesic ODE is computationally expensive and highly sensitive to initialization quality (Section 4.5). These limitations are explicitly acknowledged in their paper.
>
> **Diepeveen (2024)** proposes to learn pullback geometry using invertible neural networks, but the method has not been shown to scale beyond toy 2D datasets. Importantly, their training setup also requires computing graph-based geodesic distances—introducing the same scalability bottlenecks as Sorrenson’s method, now shifted to the training phase.
>
> In contrast, **our method** sidesteps graph-based constructions entirely. We derive closed-form expressions for key manifold maps (see Proposition 3.1), enabling efficient geodesic computation via two forward passes $\phi(x)$, $\phi(y)$ and a single inverse pass $\phi^{-1}((1 - t)\phi(x) + t\phi(y))$. This enables scalability to high-dimensional, complex distributions and additionally downstream constructions like RAEs, which are not accommodated by previous methods. We obtained high-quality geodesic paths for datasets like Blobs and MNIST, which were omitted from the original submission due to our decision to focus Section 6.2 on the RAE. We are happy to include both this discussion and the corresponding experimental results in the revised manuscript to further clarify the scalability advantages of our method.
>
> ---
>
> **Q2.)** A standard normalizing flow (NF) with a fixed identity covariance in its base distribution lacks any mechanism to distinguish important latent directions from unimportant ones, and thus cannot discover the intrinsic dimension of the data. The same limitation applies to an “isometric” NF, since its base covariance remains fixed at the identity.
>
> Consequently, the only meaningful comparison is an *anisotropic* NF **without** isometric regularization. However, in the absence of a mechanism enforcing approximate isometry, the model has no incentive to align its latent space with the data manifold—making such alignment highly unlikely to occur by chance. By contrast, isometry regularization guides the model to build a latent space approximately isometric to the data manifold, ensuring that higher-variance latent dimensions encode meaningful on-manifold variability, while collapsed-variance dimensions capture off-manifold noise. This mechanism underpins the model’s ability to uncover the intrinsic dimension of the underlying data manifold.
>
> From a *theoretical* standpoint (Theorem 4.1), the expected reconstruction error depends on how closely the diffeomorphism $\phi$ approximates an isometry, underscoring the importance of isometry regularization in the performance of RAE. *Empirically*, we observed that when the isometry weight $\lambda_{\mathrm{iso}}$ was too small—effectively an unregularized anisotropic NF—it significantly overestimated the intrinsic dimension. In contrast, setting $\lambda_{\mathrm{iso}}$ to a sufficiently large value guided $\phi$ toward an isometry and enabled the RAE to reliably and accurately uncover the true intrinsic dimensionality, in line with our theoretical analysis and intuition.
>
> We will add a remark in Section 6.2 clarifying why **both anisotropy and isometry regularization** are crucial for reliably and accurately uncovering the data’s intrinsic structure.
>
> ---
>
> **Q3.)** It is the MSE, as stated in the caption.
>
> ---
>
> ### **Other Comments Or Suggestions**
>
> We do not fully agree with the reviewer here. This connection exists if the diffeomorphism is a smooth local isometry. It is true that in the original version, we did not mention “smooth local”. We will add this in the revised version. Moreover, as we regularize for local isometry, we do have the connection to score in all of our theory and experiments.

---

> > ### Comment · Reviewer_M1sQ · 2025-04-02
> >
> > Thank you for the clarification. I now understand better the scalability benefits over the graph-based construction in __Sorrenson et al.__ and __Deepeven (2024)__. I also agree now with your perspective on calling this a score-based method, as you regularize for local isometry, which connects to the geometry based on the score function.
> > However, I believe my criticism regarding the second experiment stands -- while in theory, it may be true that only the approach that regularizes for both anisotropy and isometry can distinguish the latent dimensions, I think there is value in empirical evidence of this, which would further strengthen the arguments made for the proposed methodology.

---

> > > ### Author Response · Authors · 2025-04-05
> > >
> > > We thank the reviewer for their thoughtful follow-up and for acknowledging both the scalability benefits of our method and the relevance of using the term *score-based* in light of the local isometry regularization.
> > >
> > > We also appreciate the reviewer’s suggestion to empirically support the claim that both anisotropy and isometry regularization are necessary to reliably uncover the intrinsic dimension. We agree that including such a benchmarking analysis further strengthens our arguments, and for this reason, we have now completed the ablation study and plan to include it in the camera-ready version of the paper.
> > >
> > > As previously noted, the only meaningful ablation is between the *anisotropic flow with isometry regularization* (our proposed method) and the *anisotropic flow without regularization*. To this end, we trained the **unregularized anisotropic flow** on the same datasets used in the main paper—**Hemisphere(5,20)**, **Sinusoid(5,20)**, **blobs-10**, **blobs-20**, and **blobs-100**—all with known intrinsic dimensionality.
> > >
> > > The unregularized model significantly overestimated the intrinsic dimension in all cases:
> > > - **Hemisphere(5,20):** 19 (true: 5)
> > > - **Sinusoid(5,20):** 18 (true: 5)
> > > - **blobs-10:** 798 (true: 10)
> > > - **blobs-20:** 838 (true: 20)
> > > - **blobs-100:** 818 (true: 100)
> > >
> > > These results confirm our claim: **both anisotropy and isometry regularization** are necessary for reliably and accurately uncovering the data’s intrinsic dimensionality. We will include this empirical evidence in Section 6.2 of the updated version of the paper.
> > >
> > > Having addressed this final concern, we hope the reviewer agrees that we have fully responded to all feedback. In light of our response and the additional results provided, we kindly ask the reviewer to consider updating their evaluation.

---

### Official Review · Reviewer_dUdy · 2025-03-13

**Overall Recommendation:** 3

**Summary:**

This paper introduces a novel score-based pullback Riemannian geometry framework to extract the intrinsic geometry of data manifolds using anisotropic flows. The key contributions include:

1. Score-Based Riemannian Metric: Defines a data-driven Riemannian structure where geodesics pass through high-density regions of data.
2. Riemannian Autoencoder (RAE): Constructs a Riemannian autoencoder with error bounds to estimate the intrinsic dimension of the data manifold.
3. Integration with Normalizing Flows: Proposes modifications to normalizing flow training by incorporating isometry regularization, enabling stable manifold learning.
4. Theoretical Guarantees: Provides closed-form geodesics and error bounds for the manifold approximation.
5. Empirical Validation: Demonstrates effectiveness on synthetic and image datasets, showing superior manifold preservation and intrinsic dimensionality estimation.

**Claims And Evidence:**

I am a bit confused for some of their claims. See questions below.

**Essential References Not Discussed:**

No as far as I know.

**Experimental Designs Or Analyses:**

Yes. I think it has clear comparisons to standard normalizing flows (NF), anisotropic NF, and isometric NF.

**Methods And Evaluation Criteria:**

Yes.

**Other Comments Or Suggestions:**

Some of the claims make me a bit confused (see questions below). I would suggest make them more clear.

**Other Strengths And Weaknesses:**

Strength: Well-founded in Riemannian geometry and a novel combination of generative modeling and pullback metrics.

Weakness: Only tested on synthetic datasets and MNIST.

**Questions For Authors:**

1. Regarding the first contribution listed in section 1.1, what does it mean by 'respect' the data distribution? Can you make it formal using a rigorous math statement?

2. At the beginning of Section 3, it claims that 'the ultimate goal is ... such that geodesics always pass through the support of data probability densities'. Could you give a formal mathematical statement about this? It's not clear what it means by 'geodesics pass through the suppoart of a probability distribution'.

**Relation To Broader Scientific Literature:**

This paper builds on normalizing flows (Dinh et al., 2017) and score-based generative modeling (Song et al., 2021) and extends pullback geometry methods (Diepeveen, 2024) to a score-based setting.

**Theoretical Claims:**

I checked Proposition 3.1 and Theorem 3.3, and the derivations for geodesics and Riemannian distances look good to me.

---

> ### Author Rebuttal · Authors · 2025-03-31
>
> 1.) By “respecting” the data distribution, we mean that the geodesics induced by the metric traverse regions of high data density. For a rigorous mathematical formulation of this concept please refer to our answer to your second question. This property naturally extends to all considered manifold mappings, as they inherit this behavior from the properties of the geodesics.
>
> 2.) We thank the reviewer for bringing this up. There are different ways of formalizing this based on assumptions on the data distribution (so there is no standard way of doing this and previous work has been somewhat vague about this!). In our work this is formalized in Theorem 3.3. Right above the theorem we state:
>
> > "A direct result of Proposition 3.1 is that geodesics will pass through the support of $p(x)$ from (2), in the sense that geodesics pass through regions with higher likelihood than the end points. This can be formalized in the following result."
>
> In other words, this is the statement that tells us that the goal of data-driven Riemannian geometry has been achieved (also from this section:
>
> > "We remind the reader that the ultimate goal of data-driven Riemannian geometry on $\mathbb{R}^d$ is to construct a Riemannian structure such that geodesics always pass through the support of data probability densities.").
>
> We hope the reviewer finds the clarifications satisfactory and kindly invite them to consider updating their score in light of our response.

---

### Official Review · Reviewer_X13X · 2025-03-13

**Overall Recommendation:** 3

**Summary:**

The paper proposes a novel framework for learning the intrinsic geometry of data manifolds using pullback Riemannian metrics induced by an anisotropic normalizing flow. The key idea is to model the data manifold with a Riemannian autoencoder (RAE), where the encoder function provides a pullback metric through a strongly convex potential function. However, the clarity of presentation and theories should be heavily improved.

## update after rebuttal

I thank the authors for the rebuttal. I have re-read the response carefully and admit there are some misunderstandings in my initial review. I therefore raise my score to 3 accordingly. That said, I still have several concerns and hope the final version can be revised accordingly:

- Presentation: The author should highlight that they implicitly learn the manifold via explicitly learning $R ^d$.
- Theory: It could be argued that learning via ambient space might be restricted, as a more straightforward way is to learn the manifold directly. This should be discussed in the paper.
- Experiments: When talking about Riemannian metric learning, it should eventually recover some true Riemannian geometry, such as Frechet mean, Riemannian logarithm, etc. This is the intention of my previous comments and references, and is missing in the current version.

**Claims And Evidence:**

see wk

**Essential References Not Discussed:**

N/A

**Experimental Designs Or Analyses:**

see wk

**Methods And Evaluation Criteria:**

see wk

**Other Comments Or Suggestions:**

see wk

**Other Strengths And Weaknesses:**

**Strengths**:

Learning the Riemannian structure is an interesting topic.

**Weaknesses**:

- Several key motivations and notations lack clarification. The direct reuse of (Diepeveen, 2024) is confusing.

  - Why is Eq. (2) defined in this way, by exponential family?

  - What is $\nabla$ in Eq. (3), is it the derivative?  What is $\nabla \psi \circ \varphi(x)$?  There could be multiple understandings:  $(\nabla_x \psi) \circ \varphi$,  or $\nabla_x (\psi \circ \varphi)$, or $\nabla_x \psi \circ \varphi (x)$? The last one is the differential of $\psi \circ \varphi$ on $x$ at $x$.

  - What is $x$ in Eq. (3)? $x \in \mathcal{M} \subset R^d$ or $x \in R^d$?

  - Why is Eq. (4) defined via an SPD quadratic form? As A is SPD, why is there an inverse?

  - What is the benefit of Thm. 3.3? What are the advantages of Eq. (9) as strongly convex?

- Theoretical results seem to be questionable
  - Why there is no $\nabla$ in Eqs. (5-8)? A well-known result is that Riemannian operators under isometry are identical modulo the diffeomorphism, which is $\nabla \psi \circ \varphi$. I only see $\psi \circ \varphi$ on the left-hand side. Besides, if  $\nabla$ corresponds to the derivative, there are more issues. This $\nabla \psi \circ \varphi$ is not a common map, it is $\nabla _x \psi \circ \varphi (x)$. So, is this a diffeomorphism? only go through the proof, it seems that it is $(\nabla _x \psi) \circ \varphi (x)$. I still fail to understand why it is defined in this way. Why should we use the $(\nabla _x \psi)$? If this is the case, we can simply use $\psi$ and assume the inner product to be arbitrary (characterized by an SPD matrix). This is common in designing flat metrics induced from Euclidean spaces. Why do the authors choose a more complex presentation?

  - I fail to understand the 2nd equality in Eq. (10). I do not know why there is a $\left(D_{(\cdot)} \varphi\right)^{\top}$​.  The 1st equality is already the last one in Eq. (10)

- There is some work for learning metric tensor [c-d]. The differences and advantages of this work from the previous methods should be discussed or compared.

Experiments:

Many metrics are designed via pullback from Euclidean space, such as the Log-Euclidean/Cholesky Metric on the SPD manifold. Can the proposed method recover this synthetic scenario?


Can the proposed method handle the Riemannian submersion? This could be a harder question that the current version cannot address.
-  **Quotient**: Many geometries do not have flat structures. Many of them have a structure [a]. Many cases are orbit spaces, which are submanifolds. A prototype of the construction can be found in [Prop. A.2, b] or typical materials [e].
- **Submersion**: Many manifest as Riemannian submanifolds, whose tangent space inherits the metric from the total space. These cases are even more common than the quotient or the previous isometry cases in machine learning. Can this framework deal with this?


[a] Neural networks on Symmetric Spaces of Noncompact Type

[b] A Grassmann Manifold Handbook: Basic Geometry and Computational Aspects

[c] Riemannian Metric Learning via Optimal Transport

[d] Riemannian Metric Learning: Closer to You than You Imagine

[e] Introduction to Riemannian Manifolds

**Questions For Authors:**

see wk

**Relation To Broader Scientific Literature:**

related to manifold learning

**Theoretical Claims:**

see wk

---

> ### Author Rebuttal · Authors · 2025-03-31
>
> We thank the reviewer for their engagement. Below we address major concerns while clarifying our core contribution: A novel Riemannian geometry framework with closed-form geodesics that provably traverse high-density regions, overcoming limitations of prior data-driven approaches.
>
> - **Clarifications & Definitions**
>   - **Notation consistency:**
>     - Added explicit clarifications when citing (Diepeveen, 2024) and introduced footnotes for operator precedence (e.g., $\nabla \psi \circ \varphi \equiv (\nabla \psi) \circ \varphi$)
>   - **Equation motivations:**
>     - Eq (2): Exponential family chosen for (1) score function compatibility and (2) theoretical/algorithmic tractability with unimodal distributions
>     - Eq (4): Inverse SPD matrix preserves covariance interpretation and enables cleaner nonlinear PCA generalization
>   - **Manifold scope:** Our metric operates on all $\mathbb{R}^d$ rather than explicit submanifolds $\mathcal{M} \subset \mathbb{R}^d$
>
> - **Theoretical Contributions**
>   - **Thm 3.3 significance:** First provable guarantee that geodesics traverse density supports (Eq (9) strong convexity ensures well-posed geometry). An interpretation of the theorem is stated right above the theorem statement.
>   - Having clarified now that $\nabla \psi \circ \varphi$ should be read as $(\nabla \psi) \circ \varphi$, it might be easier to see that we get this term by the chain rule for gradients, which indeed gives that
>     $$
>     \nabla (\psi \circ \varphi)(x) = (D_x \varphi)^\top \nabla \psi(\varphi(x)).
>     $$
>
> - **Methodological Choices**
>   - $\nabla \psi$ vs. $\psi$: While $\psi \circ \varphi$ isn't a diffeomorphism, $\nabla \psi \circ \varphi$ enables pullback geometry while canceling gradient terms in our specific setting
>
> - **Comparison to [c–d]:**
>   - [c] requires predefined temporal structure; our approach is agnostic to data ordering. So it cannot be compared in our setting
>   - [d] is a review paper that acknowledges our method and does not propose any new methods (note that all relevant works cited there are discussed in our paper as well)
>
> - **Experimental Scope**
>   - **SPD/scenario recovery:** Possible in principle via diffeomorphism design, but our focus is $\mathbb{R}^d \rightarrow \mathbb{R}^d$ cases central to ML applications
>   - **Submersion limitations:** Current diffeomorphism requirement precludes dimension changes, but alternative pullback formulations remain possible. However, we feel that one should think about submersions, diffeomorphisms, and immersions as different ways of defining Riemannian geometry. We choose diffeomorphism as we get a lot for free, which is generally not the case for sub- and immersions.
>
> - **Future Directions**
>   - We see potential for applications with quotient manifolds (equivariant diffeomorphisms needed) and non-flat geometries (Diepeveen 2024 connections). While curvature/group-action extensions are compelling, they require new flow architectures beyond our present scope and the current RAE result should be generalized to account for curvature — an exciting research trajectory we hope to enable.
>
> We hope the reviewer finds the clarifications satisfactory and kindly invite them to consider updating their score in light of these revisions. (Note that we already made the mentioned changes, but are unable to upload the updated manuscript. So these changes will become visible for the camera ready version.)

---

> > ### Comment · Reviewer_X13X · 2025-04-02
> >
> > I thank the author for further clarification. However, my main concerns remain.
> >
> > > 1. "... the metric operates on $R^d$"
> >
> > Why does the Euclidean space $R^d$ need a Riemannian metric? This response is confusing to me. I first thought you were learning the geometry of an $m$-dimensional manifold $M \subset R^d$ via a diffeomorphism to $R ^m$. This case (such as the well-known figure in LLE) is quite common and rationalizes the need for learning Riemannian geometry. However, I fail to see why there are a Riemannian structure over $R^d$.
> >
> > [a] Riemann$^2$: Learning Riemannian Submanifolds from Riemannian Data
> >
> > > 2. Euclidean pullback metric recovery.  I notice that reviewer pCEZ also mentions similar comments (LEM on the SPD).
> >
> > I am a bit disappointed by this response. I **disagree** with the authors on a specific design, unless my understanding of the paper’s objective (i.e., learning data geometry via pullback) is wrong.
> >
> > I fail to see why this method cannot recover well-defined pullback metrics (from Euclidean space), as its ultimate goal is to learn a pullback metric.  As the proposed method essentially pulls back a Euclidean metric, according to my understanding.  This corresponds to the exact geometries isometric to the Euclidean space. There are many kinds of these metrics, especially in matrix manifolds. SPD is just an example I mentioned. If you insist that this method can not deal with matrices, you can even design some pullback metrics on your preferred ambient space, whose geometries are isometric to Euclidean space.
> >
> > In a word, if my understanding of this work is right, the experiments on recovering a predefined pullback metric are necessary. If the proposed method can not recover (comparatively much simpler) predefined pullback geometries, how can we guarantee it can recover more complex data geometries?

---

> > > ### Author Response · Authors · 2025-04-03
> > >
> > > We thank the reviewer for further specifying remaining concerns. Please allow us to further clarify.
> > >
> > > 1: Our work does assume that there is some lower-dimensional immersed manifold $\mathcal{M}$, but the imposed Riemannian structure is on $\mathbb{R}^d$. This setting is quite standard in machine learning settings, with data not always exactly on the manifold, but is considered to be strongly centered around it. So we indeed deviate from the setting that the reviewer had in mind, but this is for good reasons. First, it is good to highlight that our pullback geometry connects $\mathcal{M}$ and $\mathbb{R}^d$ in the following way: we aim for $\mathcal{M}$ to be a geodesic submanifold of $(\mathbb{R}^d, (\cdot, \cdot)^\varphi)$ for a suitably chosen $\varphi$. The main upshot of remetrizing $\mathbb{R}^d$ is that we get all manifold mappings for free if we choose a pullback metric (Prop 3.1) -- which gives us the tools to realize our goal. The main questions we address in the paper: (i) how to pick $\varphi$ and (ii) how to retrieve/approximate the manifold $\mathcal{M}$.
> > >
> > > We assume that we have constructed our pullback geometry from a distribution (here walking along the manifold should be interpreted as having large variance in this direction, whereas the off-manifold directions have low variance). In order for $\mathcal{M}$ to be a geodesic submanifold, we need to show that geodesics pass through the data support (which we do in Thm 3.3) and that we are able to retrieve an approximation of the data manifold (which we do in Thm 4.1). Next, having the learned the manifold in this way, geodesics under $(\mathbb{R}^d, (\cdot, \cdot)^\varphi)$ between points in $\mathcal{M}$ will stay in $\mathcal{M}$ because it is a geodesic subspace and the same holds for all other manifold mappings. This is very useful since we know all manifold mappings in closed-form because we use pullback geometry on $\mathbb{R}^d$.
> > >
> > > 2: Now having established in 1 why we want to remetrize all of space (rather than explicitly finding a diffeomorphism to a submanifold and $\mathbb{R}^m$ for $m < d$), we hope the following showcases why it is a bit subtle whether or not we are able to retrieve exact pullback metrics (and whether we'd want to in the first place!). The case of LEM metric is actually a prime example to see the subtleties.
> > >
> > > First, if we want to learn a pullback structure on $\mathbb{R}^{d\times d}$ to retrieve the set of positive definite matrices $\mathcal{P}(d)\subset \mathbb{R}^{d\times d}$ (which is in line with the setting of our paper), while insisting on learning the LEM metric, we will run into trouble. The matrix logarithm is not defined on all of $\mathbb{R}^{d\times d}$, i.e., it has a singularity at the origin and is not uniquely defined for matrices with negative eigenvalues. Nevertheless, given a data set {$\mathbf{x}^i $}$_{i=1}^N\subset \mathcal{P}(d) \subset \mathbb{R}^{d\times d}$, we expect to be able to find a submanifold with the right dimension, but we will realistically not get the LEM metric when restricting ourselves to the learned manifold. This is not an issue with the method though. Our goal is to find a pullback metric that does the job (while having closed-form manifold mappings) and not find a specific one that is not at all defined on all of the ambient space.
> > >
> > > Second, if we have information that our data points are positive definite matrices, it would make more sense to use a chart first to map them into $\mathbb{R}^m$ (with the right $m = \frac{d(d+1)}{2}$), after which we would use our framework. In this case we can use the matrix logarithm composed with projection onto the upper (or lower) triangular matrix as a chart and we only have to learn identity on $\mathbb{R}^m$, which is possible within our framework. So when extra structure is known and a specific type of metric is required, our method is flexible enough to accommodate for this. However, this is not the setting we aimed to focus on in the paper.
> > >
> > > More generally, for different settings and different matrix manifolds we feel that something similar will occur. So summarized, the answer to the question ``If the proposed method can not recover (comparatively much simpler) predefined pullback geometries, how can we guarantee it can recover more complex data geometries?'' is not as straightforward as it may seem and our goal is somewhat different from retrieving certain known pullback geometries.
> > >
> > > But to answer in short:
> > > "I fail to see why this method cannot recover well-defined pullback metrics" - if we start in euclidean space, we do recover well-defined pullback metrics. See e.g. Appendix F.1.1, where we define the metrics we aim to recover in toy examples.
> > > The statement "Possible in principle via diffeomorphism design" was referring only to setting when we are not considering the euclidean space as the base space.
> > >
> > > We believe this addresses all remaining concerns and kindly ask the reviewer to consider updating their evaluation.

---

### Official Review · Reviewer_pCEZ · 2025-03-13

**Overall Recommendation:** 4

**Summary:**

This paper proposes to construct a Riemannian structure from unimodal probability densities. Under a specific condition, the constructed pullback Riemannian structure turns out to be related to that obtained from the score function (i.e., the gradient of the log probability density with respect to data). The paper also generalizes the idea of classical PCA to the Riemannian setting, enabling the construction of Riemannian autoencoder (with error bounds on the expected reconstruction error) from unimodal probability densities. Finally, the authors show how train the density in question by adapting normalizing flow to their framework. Experimental results demonstrate the potential of the proposed method in various applications.

**Claims And Evidence:**

Yes

**Essential References Not Discussed:**

The main reference is Diepeveen, (2024) which is properly cited in the paper

**Experimental Designs Or Analyses:**

Yes

**Methods And Evaluation Criteria:**

Yes

**Other Comments Or Suggestions:**

Please see the weaknesses

**Other Strengths And Weaknesses:**

Strengs:
- This is an interesting paper that bridges works in geometric data analysis and generative modeling.
- The paper is nicely presented. The ideas are expressed in a simple and concise way which makes the paper easy to follow. I enjoy reading it.
- Experimental evaluation show the potential of the proposed method

Weaknesses:
- The paper only considers a simple setting based on unimodal probability densities. This would limit the capability of the proposed method to construct complex geometries.
- The Riemannian geometry induced by unimodal probability densities in Section 3 has close connections with several well-established Riemannian geometries such as Log-Euclidean [A] and Log-Cholesky [B]. It would be helpful to have a discussion on those connections. Currently, such a discussion is missing in the paper.

**References**

[A] V. Arsigny, P. Fillard, X. Pennec, and N. Ayache. Fast and Simple Computations on Tensors with Log-Euclidean Metrics. Technical Report RR-5584, INRIA, 2005.

[B] Lin, Z.: Riemannian Geometry of Symmetric Positive Definite Matrices via Cholesky Decomposition. SIAM Journal on Matrix Analysis and Applications 40(4), 1353–1370 (2019).

**Questions For Authors:**

Please see the weaknesses

**Relation To Broader Scientific Literature:**

The paper bases on works in geometric data analysis and generative modeling. It seems to be the first work that constructs the complete geometry of the data manifold

**Theoretical Claims:**

I read the proofs to understand the general ideas but could not follow all the details

---

> ### Author Rebuttal · Authors · 2025-03-31
>
> We thank the reviewer for their positive constructive feedback! See below for a discussion on the weaknesses:
>
> - **“The paper only considers a simple setting based on unimodal probability densities. This would limit the capability of the proposed method to construct complex geometries.”**
>
>   - This is definitely a limitation of the current method as we also mention in the paper. Having that said, for subsequent work it is important to understand the base case, which is why we focus in this paper on unimodal distributions as a first step.
>
> - **“The Riemannian geometry induced by unimodal probability densities in Section 3 has close connections with several well-established Riemannian geometries such as Log-Euclidean [A] and Log-Cholesky [B]. It would be helpful to have a discussion on those connections. Currently, such a discussion is missing in the paper.”**
>
>   - We feel that these are somewhat different things. The Log-Euclidean and Log-Cholesky metrics are pullback metrics on the space of positive definite matrices rather than on $\mathbb{R}^d$ and are not data-driven (which is the case we consider). So apart from these also being pullback metrics (and there exist many other pullback metrics on many other spaces), we don’t see a close connection as you would really do different things with both Riemannian structures. Alternatively, is the reviewer thinking about these metrics as metrics between centered normal distributions (information geometry)? In that case this is still not really related to what we are doing as we are considering metrics between elements in a distribution rather than metrics between distributions. So overall, would the reviewer maybe be more specific what this close connection is apart from the fact that they both use pullback Riemannian geometry?

---

> > ### Comment · Reviewer_pCEZ · 2025-04-07
> >
> > I thank the authors for the clarification. I thought of the connection because both metrics turn the space of PD matrices into flat spaces and the proposed method probably benefits the related litterature. I have no further questions at this time.

---

### Decision · Program_Chairs · 2025-05-01

**Decision:**

Accept (poster)

**Comment:**

The paper integrates concepts from pullback Riemannian geometry and generative models to propose a framework for data-driven Riemannian geometry that is scalable in both geometry and learning: score-based pullback Riemannian geometry. With this structure, the paper constructs an autoencoder to discover the correct data manifold dimension. The proposed framework can be used with anisotropic normalizing flows by adopting isometry regularization. Experimental results are provided demonstrating the performance of the method.

I find the paper to be very well written and interesting, and I agree with all the positive comments of the reviewers. One small drawback of the paper is that it only caters to unimodal distributions.

Given the positive reviews of the reviewer, I am happy to recommend accepting the paper. I urge authors to incorporate all the comments of the reviewers along with the rebuttal discussion in the camera-ready version of the paper.